# Chemokines act as phosphatidylserine-bound "find-me" signals in apoptotic cell clearance

**Sergio M. Pontejo** *, **Philip M. Murphy**

Molecular Signaling Section, Laboratory of Molecular Immunology, National Institute of Allergy and Infectious Diseases, National Institutes of Health, Bethesda, Maryland, United States of America

* jesussergio.martinpontejo@nih.gov

## Abstract

Removal of apoptotic cells is essential for maintenance of tissue homeostasis. Chemotactic cues termed "find-me" signals attract phagocytes toward apoptotic cells, which selectively expose the anionic phospholipid phosphatidylserine (PS) and other "eat-me" signals to distinguish healthy from apoptotic cells for phagocytosis. Blebs released by apoptotic cells can deliver find-me signals; however, the mechanism is poorly understood. Here, we demonstrate that apoptotic blebs generated in vivo from mouse thymus attract phagocytes using endogenous chemokines bound to the bleb surface. We show that chemokine binding to apoptotic cells is mediated by PS and that high affinity binding of PS and other anionic phospholipids is a general property of many but not all chemokines. Chemokines are positively charged proteins that also bind to anionic glycosaminoglycans (GAGs) on cell surfaces for presentation to leukocyte G protein–coupled receptors (GPCRs). We found that apoptotic cells down-regulate GAGs as they up-regulate PS on the cell surface and that PS-bound chemokines, unlike GAG-bound chemokines, are able to directly activate chemokine receptors. Thus, we conclude that PS-bound chemokines may serve as find-me signals on apoptotic vesicles acting at cognate chemokine receptors on leukocytes.

## Introduction

Chemokines comprise a large family of small (8 to 12 kDa) cytokines that direct leukocyte migration during homeostasis, development, and disease [1]. There are 4 chemokine subfamilies (C, CC, CXC, and CX3C) distinguished and named by the number and arrangement of conserved cysteines near the N terminus [2]. In the classic multistep model of leukocyte transendothelial migration, chemokines act by binding sequentially to 2 main classes of molecules: glycosaminoglycans (GAGs) on endothelial cells and 7-transmembrane domain G protein–coupled receptors (GPCRs) on leukocytes [3]. GAGs act as extracellular scaffolds that concentrate chemokines on the luminal surface of postcapillary venules, positioning them to attract circulating leukocytes by activating specific leukocyte GPCRs [4].

GAGs contain highly anionic sulfated polysaccharides and are thought to protect chemokines from proteases and shear forces in blood vessels [5]. Of note, GAG binding can induce the formation of high-order chemokine oligomers, and chemokine oligomers have a higher GAG-binding affinity than chemokine monomers [6]. Thus, GAG binding and

**Data Availability Statement:** All relevant data are within the paper and its Supporting Information files. Flow cytometry FCS files can be found in the FlowRepository (https://flowrepository.org/) with accession number FR-FCM-Z3NS.

**Funding:** The National Institute of Allergy and Infectious Diseases-Intramural Research Program (grant number N/A) supported SMP, PMM. The funder had no role in study design, data collection and analysis, decision to publish, or preparation of the manuscript.

**Competing interests:** The authors have declared that no competing interests exist.

**Abbreviations:** AnV, annexin V; ApoBD, apoptotic body; BLI, biolayer interferometry; BMDM, bone marrow–derived macrophage; bt, biotinylated; CL, cardiolipin; DEX, dexamethasone; DOPC, 1,2-dioleoyl-sn-glycero-3-phosphocholine; DOPE, 1,2-dioleoyl-sn-glycero-3-phosphoethanolamine; DOPEbiot, biotinylated DOPE; DOPS, 1,2-dioleoyl-sn-glycero-3-phospho-L-serine; DSPE-PEGbiot, 1,2-distearoyl-sn-glycero-3-phosphoethanolamine-N-[biotinyl(polyethylene glycol)-2000]; FACS, fluorescence-activated cell sorting; FSC, forward scatter; GAG, glycosaminoglycan; GPCR, G protein–coupled receptor; hpf, high-power field; i.p, intraperitoneal; MFG-E8, milk fat globule-epidermal growth factor 8; MM1, MonoMac-1; MV, microvesicle; NIAID, National Institute of Allergy and Infectious Diseases; oxLDL, oxidized LDL; PC, phosphatidylcholine; PE, phosphatidylethanolamine; PI, propidium iodide; PMSF, phenylmethylsulfonyl fluoride; PS, phosphatidylserine; RFU, relative fluorescence units; R-PE, R-Phycoerythrin; SN, supernatant; SR-PSOX, scavenger receptor for PS and oxLDL; SSC, side scatter; UV, ultraviolet; VDAC1, voltage-dependent anion-selective channel 1.

oligomerization may be mutually reinforcing processes that promote formation of haptotactic gradients and presentation of chemokines to leukocytes. However, since GAG binding has been found to interfere with the receptor-binding activity of many chemokines [5], it has been recently proposed that chemokines are trapped, stabilized, and concentrated by GAGs, but they must be released in order to activate cognate receptors on leukocytes [7].

Several other types of molecules also bind chemokines; however, the functional significance is not well defined, and the interactions tend not to be shared in a class sense within or across chemokine subfamilies. A particularly interesting example is CXCL16, which binds the GPCR CXCR6 as well as phosphatidylserine (PS) and oxidized LDL (oxLDL) [8–10]. The latter activities suggested that CXCL16 might function as a scavenger receptor, particularly since it is a multimodular protein containing a transmembrane domain, one of only 2 chemokines with this type of structure [11]. In fact, CXCL16 was originally named SR-PSOX or "scavenger receptor for PS and oxLDL" [12]. While the role of the CXCL16–oxLDL interaction in atherosclerosis has been extensively studied [13–15], the biological relevance of PS as a ligand for CXCL16 remains poorly understood. Moreover, although oxLDL has been shown to bind other chemokines [16], to date, CXCL16 is the only reported PS-binding chemokine.

PS is an anionic phospholipid that is normally concealed within the inner leaflet of the plasma membrane of living cells [17]. As cells undergo apoptosis, PS relocates to the outer leaflet of the plasma membrane where it serves as a marker of apoptotic cell death and as an "eat-me" signal promoting recognition and engulfment of apoptotic cells by phagocytes [18–20]. Conversely, chemotactic "find-me" signals, including the chemokine CX3CL1, the nucleotides ATP and UTP, lysophosphatidylcholine, and sphingosine 1-phosphate, guide phagocytes toward apoptotic cells [21–25]. In addition, apoptotic blebs can be shed as PS-exposing extracellular vesicles able to recruit phagocytes [26–28]. Cellular receptors for the 4 classical find-me signals have been identified, but how apoptotic blebs exert their chemotactic find-me signal action on phagocytes has remained undefined. Here, we provide evidence that chemokine binding to PS on apoptotic blebs can mediate this activity.

# Results

## Many human chemokines bind anionic phospholipids including phosphatidylserine

We first conducted a simple screen for chemokine binding to phospholipids using arrays of 15 different phospholipids that are found in biological membranes (left panel in Fig 1A). Of the 10 human chemokines tested, CCL3, CXCL6, CCL5, and CXCL8 displayed very weak or no binding capacity to the array, whereas the other 6 chemokines bound selectively to multiple anionic phospholipids (Fig 1A). The binding pattern was similar for CCL11, CCL21, CXCL9, and CXCL11, with strong binding to PS and cardiolipin (CL) and weaker interaction with the other anionic phospholipids on the array (Fig 1A). CXCL3 also bound to PS and CL; however, its strongest binding was to sulfatide. CCL20 appeared to be CL selective (Fig 1A). The preference of all 6 of these chemokines for PS and CL over the more highly anionic phosphoinositides suggests some degree of specificity and that these interactions are not solely driven by charge.

Since CL is not found in the plasma membrane of mammalian cells, we focused on the chemokine–PS interaction. To confirm that chemokines can interact with PS in the context of a phospholipid bilayer, we studied by ELISA chemokine binding to PS-containing liposomes. For this, we immobilized "DOPC" liposomes (containing 1,2-dioleoyl-sn-glycero-3-phosphocholine [DOPC] only) or "DOPS" liposomes (containing 30% 1,2-dioleoyl-sn-glycero-3-phospho-L-serine [DOPS] and 65% DOPC), both containing trace amounts (5%) of

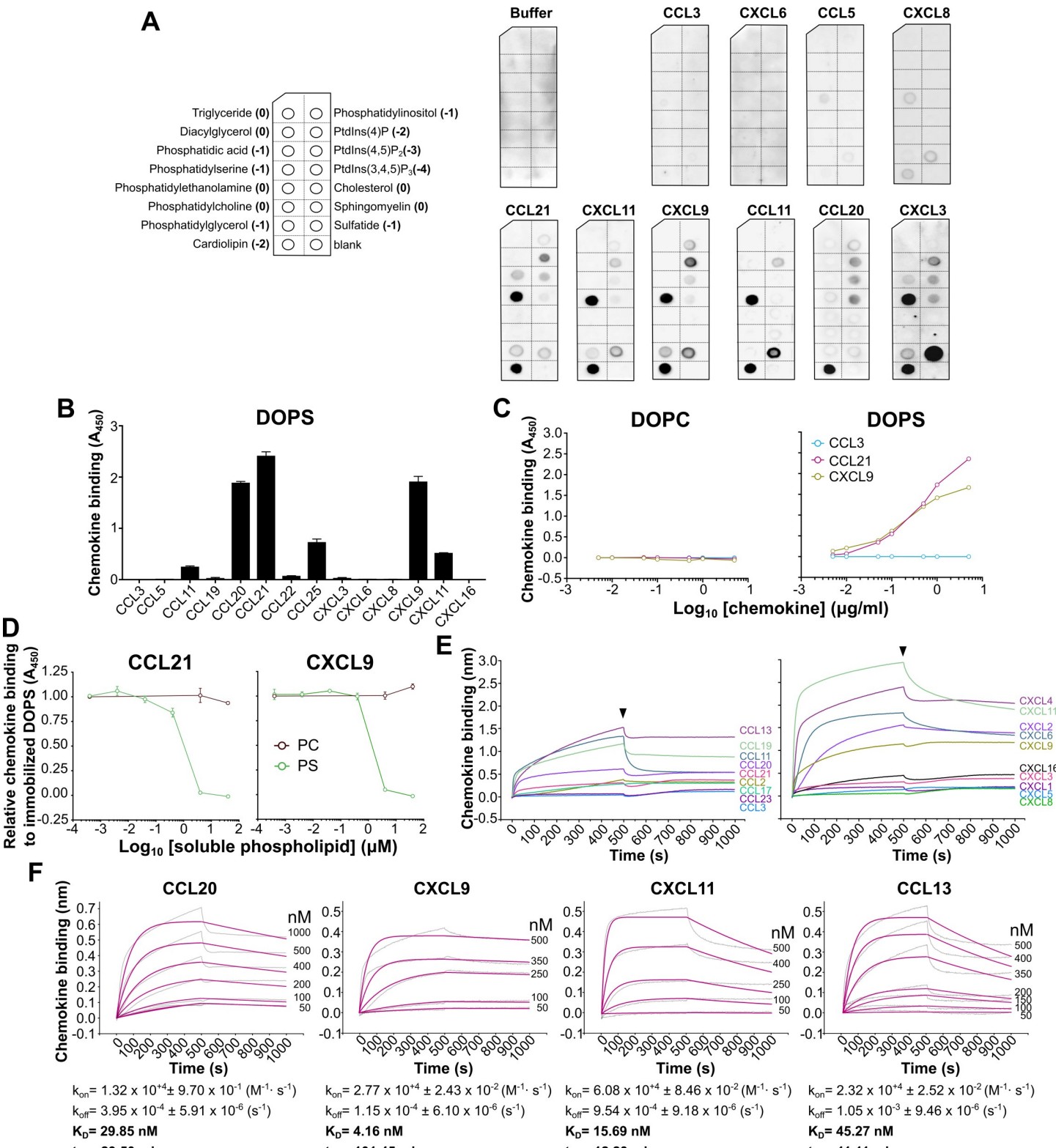

**Fig 1. Chemokines bind to anionic phospholipids.** The phospholipid-binding activity of chemokines was studied by protein–lipid overlay (**A**), ELISA (**B–D**), and BLI (**E, F**). (A) Arrays spotted with 15 different phospholipids (left panel; in parentheses, net charge of each phospholipid) were incubated with buffer or 0.1 μg/ml of the indicated chemokines. Bound chemokine was detected with specific antibodies. Results are representative of 2–3 experiments for each chemokine. (B) Human chemokines indicated on the x-axis were incubated at 1 μg/ml in wells containing immobilized DOPS liposomes. Bound chemokine was detected with specific antibodies for each chemokine,

and the absorbance at 450 nm ($A_{450}$) was calculated in a microplate reader. Data are presented as mean ± SD chemokine binding of triplicate $A_{450}$ determinations in one experiment representative of 2 independent experiments. (C) Increasing concentrations of CCL3, CCL21, or CXCL9 were incubated in wells containing immobilized DOPC (left) or DOPS (right) liposomes. Bound chemokine was detected as in B. Data are presented as mean ± SD of triplicates from one experiment representative of 3 independent experiments. (D) Increasing concentrations of the soluble phospholipids indicated in the inset of the right panel were preincubated with 0.5 μg/ml of CCL21 (left) or CXCL9 (right). The chemokine–lipid mix was then added into wells containing immobilized DOPS liposomes, and liposome-bound chemokine was detected as in B. Chemokine binding data are presented as the mean ± SD of triplicate $A_{450}$ determinations relative to the $A_{450}$ recorded in the absence of competing phospholipid and are representative of 3 independent experiments. (E) Chemokine–PS binding screen by BLI. Biosensors immobilized with DOPS liposomes were incubated with 1 μM of the indicated human CC (left panel) and CXC (right panel) chemokines color-coded on the right side of each curve. The binding response in nm (y-axis) over time (x-axis) for each chemokine is shown. The binding of each chemokine to biosensors coated with DOPC liposomes was used as reference and subtracted from the corresponding binding curve. Black arrowheads point to the beginning of the dissociation phase. (F) Kinetic analysis. Selected chemokines (indicated above each panel) were incubated at the concentrations indicated to the right of each curve with biosensors coated with DOPS liposomes. Binding of each chemokine concentration to reference DOPC biosensors was subtracted. The obtained binding curves (gray) were globally fitted to a 1:1 Langmuir model (magenta curves), and the association ($k_{on}$), dissociation ($k_{off}$), and affinity ($K_D$) constants were calculated. The half-life ($t_{1/2}$) for each interaction was calculated as $t_{1/2} = ln(2)/k_{off}$. The underlying numerical values for the panels displaying summary numerical data can be found in S1 Data. BLI, biolayer interferometry.

biotinylated 1,2-dioleoyl-sn-glycero-3-phosphoethanolamine (DOPEbiot), onto streptavidin-coated plates. Except where indicated, all the phospholipids in this study were used in the form DOPC, DOPS and DOPE, but for clarity purposes, hereon, we will refer to these phospholipids simply as PC, PS, and PE, respectively, whereas the terms "DOPC" and "DOPS" will be used to designate the 2 different types of liposomes used in our experiments. Consistent with the results in Fig 1A, CCL11, CXCL11, CCL21, and CXCL9, but not CCL3, CCL5, CXCL6, and CXCL8 bound to DOPS liposomes to some extent (Fig 1B). Of note, since different primary antibodies were used to detect each chemokine, direct quantitative comparisons cannot be made from this ELISA-based lipid-binding experiment. Importantly, CXCL9 and CCL21, but not CCL3, bound to DOPS but not to DOPC liposomes in a dose-dependent manner (Fig 1C), and increasing concentrations of soluble PS ($IC_{50} \approx 1$ μM) but not PC blocked their binding to DOPS liposome-coated plates (Fig 1D). These results confirmed the PS specificity of these chemokines. On the other hand, as shown in Fig 1B, CXCL3 was not able to recognize PS in liposomes, and CCL20, which appeared to be CL specific by lipid array (Fig 1A), displayed a strong binding to DOPS liposomes by ELISA. Besides the different disposition of the lipids (spotted purified lipids versus liposomes), differences in the experimental conditions may account for these few discrepancies between lipid array- and ELISA-based binding experiments. Also, unexpectedly, we did not detect binding of CXCL16 to DOPS liposomes by ELISA (Fig 1B).

To directly and reliably compare the PS-binding activity of different chemokines without the need of exogenous protein tags or detection antibodies, we next analyzed chemokine–PS binding by biolayer interferometry (BLI). To validate BLI as a method to study liposome–protein interactions, we first analyzed binding of milk fat globule-epidermal growth factor 8 (MFG-E8 or lactadherin), a well-known PS-binding protein [29]. As shown in S1 Fig, MFG-E8 bound to biosensors immobilized with DOPS liposomes, but not with DOPC liposomes. Using this technique, we analyzed PS binding for 21 human chemokines (S1 Table). As shown in Fig 1E and S1 Table, CXCL16, but also many other chemokines from the CC and CXC subfamilies, including CXCL3, whose binding was not detectable by ELISA (Fig 1B), bound to PS-containing liposomes. Consistent with the ELISA and phospholipid array results, CCL3 and CXCL8 did not interact with DOPS liposomes by BLI, but CXCL6, which lacked PS-binding activity in previous experiments, strongly bound DOPS liposomes by BLI (Fig 1E). Besides the numerous differences in the experimental conditions, the few discrepancies observed between BLI and the antibody-based binding assays might be due to the inability of certain anti-chemokine antibodies to detect their target chemokine complexed with lipid. To test this hypothesis, we analyzed by BLI the binding of CCL21 and CXCL9, which consistently bound to PS in all experimental settings, and of CXCL6 and CXCL3, which only bound DOPS liposomes by BLI, to their corresponding detection antibodies in the presence of DOPS

liposomes. In BLI experiments, at the same concentration, large analytes (chemokine–liposome complex) are expected to cause larger binding responses than small molecules (chemokine alone). As shown in S2 Fig, binding of CCL21 and CXCL9 to BLI biosensors immobilized with the appropriate anti-chemokine antibody was increased in the presence of DOPS liposomes but not DOPC liposomes. In contrast, this signal increase was not observed for CXCL6 and CXCL3 (S2 Fig), suggesting that the corresponding antibodies are unable to detect these chemokines in a complex with liposomes. This may explain why the PS binding of CXCL6 and CXCL3 was not detectable by ELISA. Also, importantly, the interaction of most chemokines with PS appeared to be very stable, as indicated by a slow dissociation phase (Fig 1E). We determined the kinetic parameters for 4 selected chemokines and obtained binding affinity constants ($K_D$) ranging from 4 to 45 nM and half-life ($t_{1/2}$) values between 10 and 100 minutes (Fig 1F). This binding affinity is near the range of the affinity of the MFG-E8–PS interaction ($K_D$ = 3.3 nM) [30]. These binding and kinetic analyses were performed with liposomes containing 30% PS, but we found that the PS detection limit of CCL20, CCL19, and CXCL11 was in the 5% to 10% range (S3 Fig). Taken together, these results demonstrate that CXCL16 is not exceptional, since PS is a ligand for many human chemokines.

## Anionic phospholipids induce chemokine oligomerization

Chemokines bound to cell surface GAGs oligomerize and form haptotactic gradients. We tested whether PS or other phospholipids were also capable of inducing chemokine oligomerization. Using cross-linking assays we found that, consistent with their lipid-binding specificity, the oligomerization of CCL11, CXCL11, CXCL9, and CCL21 was enhanced in the presence of anionic PS or CL, but not in the presence of zwitterionic phospholipids (phosphatidylethanolamine [PE] or PC) (Fig 2). In contrast, consistent with the fact that CCL3 does not bind anionic phospholipids, the CCL3 oligomers detected in the presence of PS or CL did not differ from the oligomers cross-linked in the absence of phospholipid (Lane "−") (Fig 2). These results further confirmed the interaction of chemokines with anionic phospholipids and support PS as a new potential substrate for the formation of chemokine oligomers and haptotactic gradients.

## Chemokine binding to phosphatidylserine does not interfere with chemotactic activity

Next, we investigated whether chemokine binding to DOPS liposomes affects chemokine bioactivity. For this, we evaluated 2 PS-binding chemokines, CCL20 and CCL21, and included the

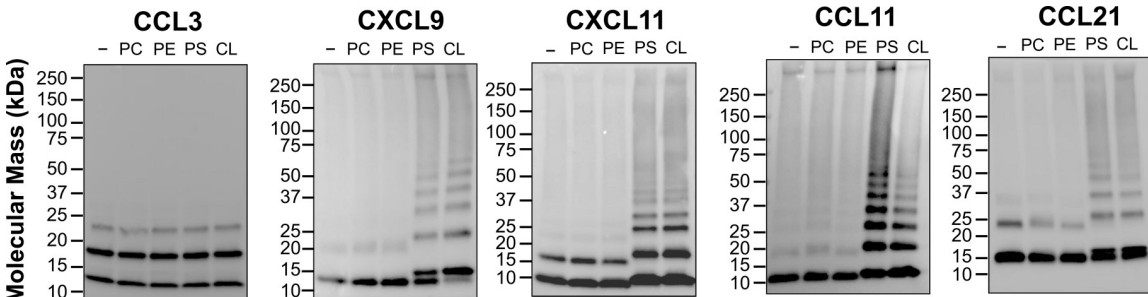

**Fig 2. Anionic phospholipids induce oligomerization of PS-binding chemokines.** The chemokines (50 ng) indicated above each panel were incubated in the absence or presence of the phospholipids (1:8, chemokine:lipid molar ratio) indicated above each lane (−, no lipid) with the cross-linker BS₃. Samples were analyzed by SDS-PAGE and immunoblot using specific anti-chemokine antibodies. Molecular mass markers are shown in kDa on the left of each panel. Data are representative of 2 independent experiments for each chemokine. CL, cardiolipin; PC, phosphatidylcholine; PE, phosphatidylethanolamine; PS, phosphatidylserine.

non-binder CCL3 as a negative control. Each chemokine was tested for its ability to chemoattract L1.2 mouse B cell lymphoma cells expressing the cognate mouse receptors Ccr6, Ccr7, and Ccr1, respectively, in the presence of increasing doses of DOPS or DOPC liposomes. As shown in the left column of Fig 3A, 1 nM CCL20, CCL21, and CCL3 induced potent chemotactic responses that remained unaltered even in the presence of a $10^5$-fold molar excess of DOPS liposomes. As expected, DOPC liposomes did not inhibit any of these chemokines either (Fig 3A). Importantly, DOPC and DOPS liposomes alone did not induce cell migration of Ccr1-, Ccr6-, or Ccr7-expressing L1.2 cells (panel A in S4 Fig). In order to confirm that the chemokines bound to the liposomes under these experimental conditions, the presence of liposome-bound chemokine was analyzed by ELISA. Using the DOPEbiot introduced in our liposomes, a small aliquot of each chemokine–liposome mix prepared for the chemotaxis assays was immobilized on streptavidin-coated plates. Consistent with their PS-binding activity, CCL20 and CCL21 but not CCL3 were detected only when mixed with DOPS liposomes (right column in Fig 3A). These findings indicate that high doses of PS do not impair the activity of PS-binding chemokines, but they do not directly prove whether or not a chemokine–liposome complex can activate the receptor. To address this question, exploiting the biotin present in all our liposomes, we analyzed the chemotactic activity of the CCL20–DOPS mix after depletion of the chemokine–liposome complexes by pull-down with Strep-Tactin (streptavidin analog)-coated beads. As shown in panel B of S4 Fig, the levels of soluble CCL20–DOPS complexes detected by ELISA in streptavidin-coated plates were markedly reduced after pull-down with Strep-Tactin-beads, which resulted in at least a 40% reduction of cell migration compared to pull-down supernatants (SNs) from samples containing CCL20 alone or mixed with DOPC liposomes (panel C in S4 Fig). In contrast, pull-down did not affect migration induced by CCL3 in the presence of DOPS liposomes (panel C in S4 Fig). These results suggest that the CCL20–DOPS complex contributes to the overall cell migration observed in Fig 3A, and, therefore, at least in the case of CCL20, PS binding appears to be compatible with the chemotactic activity of the chemokine. This observation contrasts with the inhibitory effect reported for soluble GAGs on several chemokines and a recently proposed model suggesting that while GAGs present chemokines on endothelial cells, they may need to be released to act at leukocyte GPCRs [7,31]. In fact, we found that while a $10^4$-fold molar excess of DOPS or DOPC liposomes did not alter the chemotactic potency or the efficiency of CCL3, CCL20, and CCL21, a $10^3$-fold molar excess of the soluble GAG heparin sufficed to nearly neutralize CCL20 and CCL21 (Fig 3B). In contrast, heparin did not block CCL3 (Fig 3B), probably due to the known low binding affinity of this chemokine for GAGs [6,32]. Thus, PS-binding chemokines engage PS by a molecular mechanism that, unlike GAG engagement, permits binding of the chemokine to cellular receptors.

### Phosphatidylserine-binding chemokines interact with the surface of apoptotic blebs and necrotic and apoptotic cells by a mechanism involving membrane-exposed phosphatidylserine

Since PS is normally located on the inner leaflet of the plasma membrane of living cells but flips to the outer leaflet of the plasma membrane of cells undergoing apoptosis [17], cell death is an obvious biological process in which PS binding by chemokines may be functionally relevant. This property of PS is exploited for detection of apoptotic cells by staining with the PS-binding protein annexin V (AnV) [33]. Similarly, apoptotic blebs released by dying cells also expose PS on their surface [34]. Therefore, we studied whether chemokines might interact with the surfaces of apoptotic blebs, dying CHO cells, and dying primary mouse thymocytes.

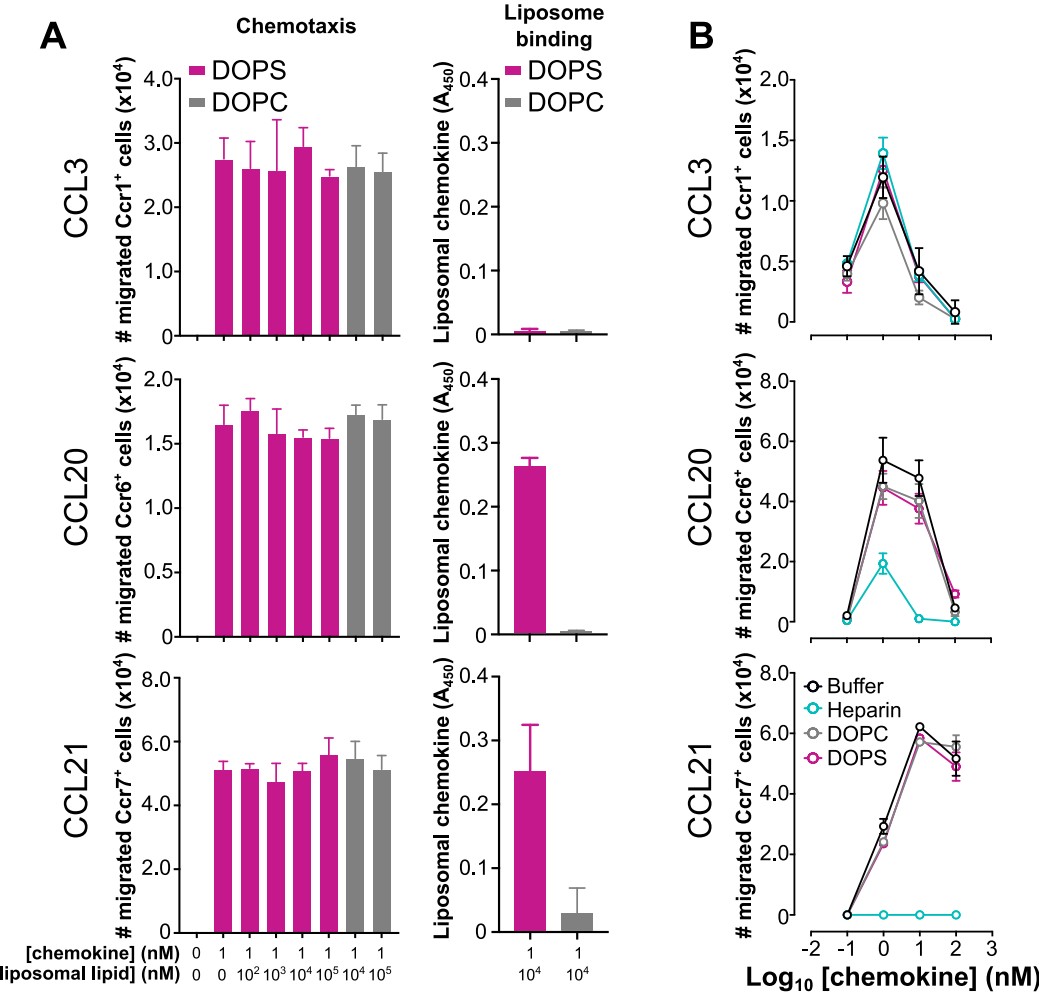

**Fig 3. Chemokine interaction with liposomal PS does not affect chemotactic activity. (A)** DOPS liposomes do not impair chemokine chemotactic activity. Left column: "Chemotaxis." The chemokines indicated on the left side of each graph row were preincubated for 30 minutes at room temperature with biotinylated DOPS (magenta) or DOPC (gray) liposomes at the concentrations indicated below the x-axis. Then, chemotactic activity of the mixture was tested using L1.2 reporter cells, stably expressing the mouse receptors Ccr1 (for CCL3), Ccr6 (for CCL20), and Ccr7 (for CCL21) as indicated on the y-axis. Bars correspond to the total number of migrated cells after 3–4 hours at 37°C for each chemokine–liposome mix or buffer alone (0:0, chemokine:liposome). Right column: "Liposome binding." CCL20 and CCL21 but not CCL3, bound to DOPS liposomes under the experimental conditions used for the chemotaxis assays. Liposome-bound chemokine was detected by ELISA. A total of 50 μl of the $1:10^4$ chemokine:liposome molar ratio mix of each chemokine with biotinylated DOPS or DOPC liposomes (as indicated in the inset of the top graph) were incubated in streptavidin-coated wells. After washing, liposome-bound chemokine was detected by specific antibodies, and $A_{450}$ was calculated in a microplate reader. All data are presented as mean ± SD of triplicates from one experiment representative of 3 independent experiments. **(B)** Anionic GAGs but not PS-containing liposomes inhibit the chemotactic activity of CCL20 and CCL21. Increasing concentrations (0.1–100 nM) of CCL3, CCL20, and CCL21 (as indicated on the left side of each graph) were preincubated for 30 minutes at room temperature with buffer alone (black), or a $10^4$-fold molar excess of DOPC liposomes (gray) or DOPS liposomes (magenta), or a $10^3$-fold molar excess of heparin (cyan), as indicated in the inset of the bottom graph. Chemotactic activity for each condition is presented as the mean ± SD number of migrated cells of triplicates from one experiment representative of 2–3 independent experiments. The underlying numerical values for the panels displaying summary numerical data can be found in S1 Data. GAG, glycosaminoglycan; PS, phosphatidylserine.

In order to obviate chemokine-binding interference by cell surface GAGs, we initially used GAG-deficient CHO-745 cells rendered apoptotic by ultraviolet (UV) light irradiation. By fluorescence-activated cell sorting (FACS) analysis, we observed 2 distinct cell populations,

forward scatter high (FSC$^{hi}$) and low (FSC$^{lo}$), in both mock-treated and UV-irradiated cells, with the percentage of FSC$^{lo}$ cells being markedly increased after exposure to UV light (Mock FSC$^{lo}$, 10.3%; UV FSC$^{lo}$, 34.7%) (Fig 4A). In addition, a high percentage (19.8%) of small side scatter low (SSC$^{lo}$) events, most likely corresponding to apoptotic blebs, was detected in UV-treated CHO-745 cells, whereas this population was nearly undetectable (0.6%) in mock-treated cells (Fig 4A). We found that most FSC$^{lo}$ events were AnV$^+$ and propidium iodide$^+$ (PI$^+$) (necrotic cells), whereas FSC$^{hi}$ cells were AnV$^-$ PI$^-$ (live cells) (Fig 4A). Therefore, for subsequent chemokine binding experiments, we relied on this FSC profile to distinguish live and necrotic cells without the need for AnV staining, which might interfere with chemokine binding to cell surface–exposed PS. We analyzed the binding of 4 biotinylated proteins, AnV as positive control, CCL3 as negative control, and 2 PS-binding chemokines, CCL11 and CXCL11, to 3 sources of target membranes: FSC$^{lo}$ necrotic cells (surface PS positive), FSC$^{hi}$ live cells (surface PS negative), and apoptotic blebs (surface PS positive). As mentioned above, although UV irradiation clearly increased the frequency of necrotic CHO-745 cells, a small but measurable background % of AnV$^+$ PI$^+$ necrotic cells was consistently found in mock-treated samples (Fig 4A). However, it is important to note that AnV binding to mock-treated spontaneous necrotic cells was similar to that detected on UV-irradiated necrotic cells (compare the orange AnV-bt histogram in the "Mock" panel with the orange AnV-bt histogram in the "UV" panel of Fig 4B), indicating that these 2 types of necrotic cells were equivalent in terms of PS exposure on the plasma membrane. Consistent with this and with their PS-binding activity, CCL11 and CXCL11, but not CCL3, displayed binding to necrotic cells that was ≥2-log stronger than to live cells regardless of the treatment (Fig 4B). To explore whether these chemokines were also able to interact with apoptotic blebs, we analyzed their binding to an SSC$^{lo}$ population found only in UV-irradiated samples after gating cellular debris with extremely low SSC values out of the analysis (Fig 4C, left column). As shown in Fig 4C, the PS-binding proteins CCL11, CXCL11, and AnV, but not the PS-non-binder CCL3, bound to the surface of apoptotic blebs. In addition to these commercial synthetic non-glycosylated chemokines, we confirmed that glycosylated CCL21 and CXCL9 produced in Expi293F cells were also able to bind to PS-containing liposomes as well as to necrotic CHO-745 cells (S5 Fig). To rule out the possibility that the cognate cellular receptors of CCL11, CXCL9, CXCL11, and CCL21 could be mediating their binding to apoptotic cells, we analyzed the expression of CCR3 (receptor for CCL11), CXCR3 (receptor for CXCL9 and CXCL11), and CCR7 (receptor for CCL21) in CHO-745 cells. As shown in panel A of S6 Fig, no significant expression of these receptors was detected in either mock-treated or AnV$^-$ and AnV$^+$ populations in UV-irradiated cells. Thus, we conclude that PS-binding chemokines interact with necrotic CHO-745 cells and blebs in a GAG- and chemokine receptor–independent manner.

UV irradiation of CHO-745 cells failed to produce AnV$^+$ PI$^-$ apoptotic cells. To study the binding of chemokines to early apoptotic cells while at the same time expanding our analysis to primary cells, we induced apoptosis in mouse thymocytes by ex vivo treatment with dexamethasone (DEX), a well-characterized model of apoptosis in primary lymphocytes [35]. To distinguish live cells from apoptotic cells without using AnV, which might interfere with chemokine binding to PS, we used YO-PRO, which selectively permeates apoptotic cells in a PS-independent manner [36]. As shown in Fig 4D, 4 hours after DEX treatment, YO-PRO and AnV detected similar levels of early apoptotic thymocytes (53.5% and 50.4%, respectively). Then, we studied the binding of several biotinylated chemokines, the PS-non-binder CCL3 as negative control and 4 PS-binding chemokines, CCL2, CCL11, CXCL11, and CCL20, to live (YO-PRO$^-$ PI$^-$) and early apoptotic (YO-PRO$^+$ PI$^-$) thymocytes. As shown in Fig 4E, CCL2, CCL11, CXCL11, and CCL20 bound apoptotic cells to different degrees (CXCL11 > CCL20 = CCL11 > CCL2), whereas no binding was detected for CCL3. Importantly, in the presence of

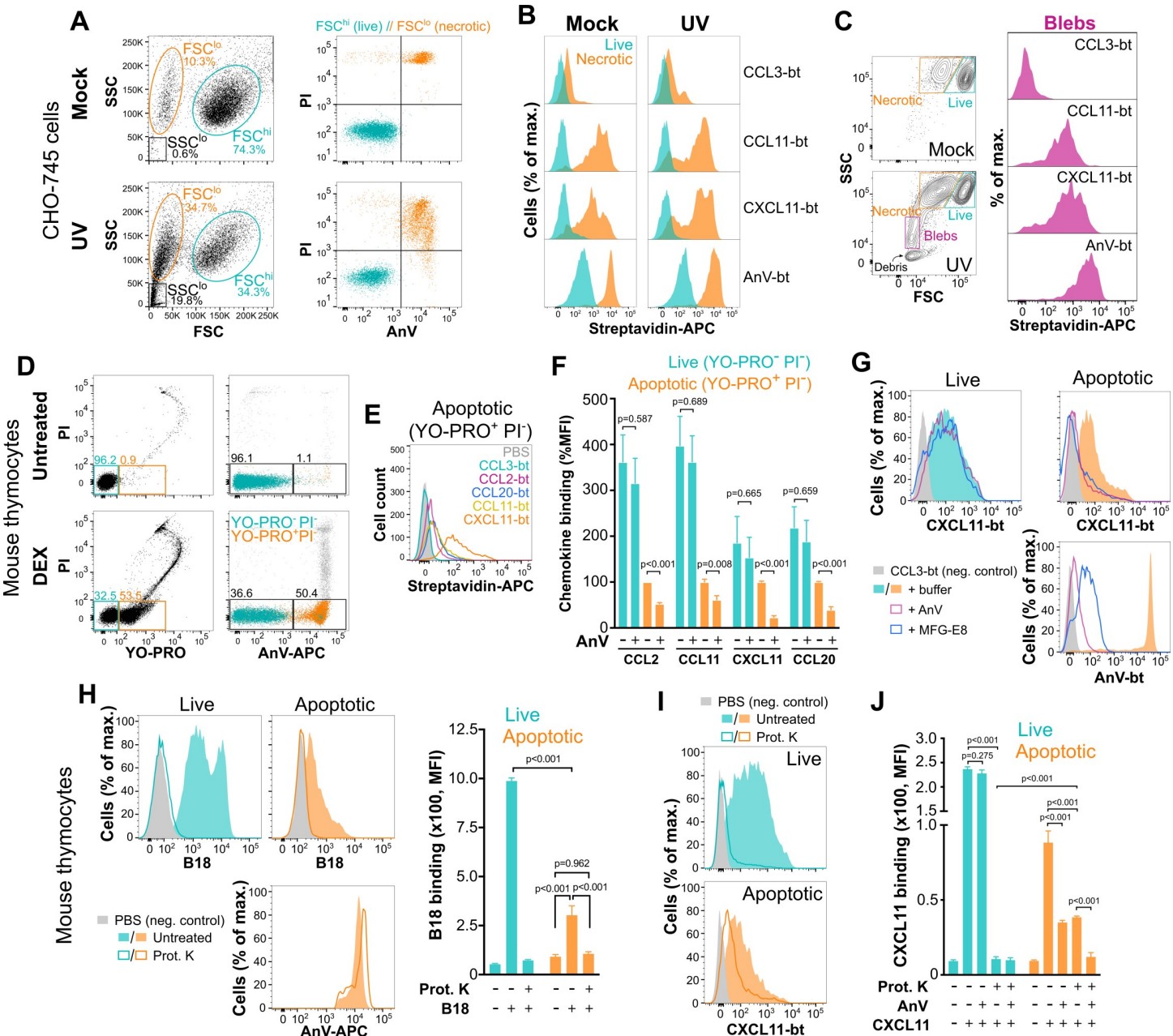

**Fig 4. PS-binding chemokines exploit membrane-exposed PS to interact with the surface of dying cells and apoptotic blebs.** As indicated on the left side of each panel row, panels A–C and D–J correspond to analysis based on GAG-deficient CHO-745 cells and mouse thymocytes, respectively. **(A)** FACS analysis and gating of mock- and UV-irradiated CHO-745 cells. FSC–SSC dot plots are shown in the left column. Gates for FSC$^{lo}$ necrotic cells, FSC$^{hi}$ live cells, and SSC$^{lo}$ blebs are shown in orange, cyan, and black, respectively. In the right column, dot plots for AnV and PI staining of FSC$^{lo}$ necrotic and FSC$^{hi}$ live populations are colored corresponding to their gates in the left column. **(B, C)** Binding of the indicated bt proteins to necrotic (orange) and live (cyan) cells from mock and UV-irradiated CHO-745 cells in B, or to blebs (magenta) in C, detected with streptavidin-APC (x-axis). In C, left column, are log SSC/FSC contour plots (5% level plus outliers) of mock- and UV-treated cells, showing the gates of live (cyan) and necrotic (orange) CHO-745 cells, and the gate of blebs (magenta) included in the binding analysis in the right column. **(D)** PI/YO-PRO (left column) and PI/AnV (right column) dot plots of freshly isolated mouse thymocytes (untreated) or incubated ex vivo with 1 μM DEX. In the left column, gates for YO-PRO$^-$PI$^-$ (blue, live cells) and YO-PRO$^+$PI$^-$ (orange, apoptotic cells) cells are shown, and their PI/AnV profile is represented in the right column with events colored according to their PI/YO-PRO staining. Numbers above each gate indicate % of total events. **(E)** PS-binding chemokines bind apoptotic thymocytes. Representative FACS histograms of the binding of PBS alone or the bt chemokines color-coded in the inset to early apoptotic cells (YO-PRO$^+$ PI$^-$) from DEX-treated mouse thymocytes. **(F)** AnV partially competes chemokine binding to apoptotic thymocytes. Quantification of the MFI of the binding of the indicated bt chemokines to live cells (YO-PRO$^-$PI$^-$, cyan) or early apoptotic cells (YO-PRO$^+$PI$^-$, orange) from DEX-treated mouse thymocytes in the absence or presence of unlabeled AnV as indicated on the x-axis. All binding data points ($n = 4$–6) generated for each chemokine in 3 independent experiments were combined and represented as % MFI relative to the binding of each chemokine to apoptotic thymocytes in the absence of AnV, which was set at 100%. Bars represent mean ± SEM % MFI. *p*-Values from multiple *t* tests with Holm–Sidak correction for multiple comparisons are indicated. **(G)** Binding of bt CCL3 (solid gray), or CXCL11 and AnV (as indicated on the x-axis of each graph) in the presence of buffer (solid-

colored histograms) or unlabeled AnV (open magenta) or MFG-E8 (open blue), as indicated in the legend, to live or early apoptotic thymocytes as indicated above each graph column. In E, F, and G, binding of bt proteins was detected with streptavidin-APC. **(H)** Cell surface GAGs are severely depleted in apoptotic thymocytes. Analysis of cell surface GAGs based on the cell-binding activity of the specific GAG-binding protein B18 on DEX-treated thymocytes. The FACS graphs on the left show the binding of PBS alone (solid gray), recombinant His-tagged B18 (200 nM) and AnV-APC (as indicated on the x-axis of each graph) to live or apoptotic thymocytes (as indicated above each graph column) treated (open-colored histograms) or not (solid-colored histograms) with Prot. K to remove cell surface GAGs. B18 binding was detected with an anti-His mAb. The bar graph on the right shows the quantification of the binding of B18 to live (cyan) and apoptotic (orange) thymocytes treated or not with Prot. K as indicated below the x-axis. Bars represent the mean ± SD MFI of triplicates. Results are from one experiment representative of 2 independent experiments. **(I)** The PS-binding chemokine CXCL11 binds to GAG-free apoptotic thymocytes. FACS graphs showing the binding of PBS alone (solid gray) or bt CXCL11 (500 nM) to live (cyan) or apoptotic (orange) DEX-treated thymocytes treated (open-colored histograms) or not (solid-colored histograms) with Prot. K to remove cell surface GAGs. **(J)** Quantification of the binding of CXCL11 to live (cyan) and apoptotic (orange) thymocytes treated or not with Prot. K and in the presence or absence of unlabeled AnV as indicated below the x-axis. Bars represent the mean ± SD MFI of triplicates. Results are from one experiment representative of 2 independent experiments. Chemokine binding in I and J was detected as in panels E, F, and G. *p*-Values in panels H and J are from a 2-way ANOVA test with Tukey correction for multiple comparisons. The underlying numerical values for the panels displaying summary numerical data can be found in S1 Data. AnV, annexin V; bt, biotinylated; DEX, dexamethasone; FACS, fluorescence-activated cell sorting; FSC, forward scatter; GAG, glycosaminoglycan; MFI, median fluorescence intensity; PI, propidium iodide; Prot. K, proteinase K; PS, phosphatidylserine; SSC, side scatter; UV, ultraviolet.

unlabeled AnV, the binding of these chemokines to apoptotic cells was reduced by at least 40%, whereas no significant competition of chemokine binding to live thymocytes by unlabeled AnV was observed (Fig 4F). Furthermore, not only AnV, but also unlabeled MFG-E8, 2 specific PS-binding proteins, which, as expected, effectively competed the binding of biotinylated AnV to apoptotic cells, reduced the interaction of CXCL11 with apoptotic thymocytes without affecting chemokine binding to live cells (Fig 4G). These results demonstrate that CXCL11 binding to apoptotic but not live thymocytes is at least partially mediated by surface membrane-exposed PS.

The partial resistance of chemokine–apoptotic cell binding to AnV competition observed in Fig 4F might be explained by the presence of cell surface GAGs or the background expression of cognate chemokine receptors in mouse thymocytes. To address these possibilities, we first analyzed by FACS the expression of Ccr2 (receptor for CCL2), Ccr3 (CCL11), Ccr6 (CCL20), and Cxcr3 (CXCL11) in DEX-treated thymocytes. Interestingly, we found strong Ccr3 expression selectively in apoptotic thymocytes, whereas only very modest to non-detectable levels of the other GPCRs were observed in either live or apoptotic cells (panel B in S6 Fig), indicating that GPCR-mediated interference in the AnV competition assay was unlikely except in the case of CCL11 binding to apoptotic cells. In addition, it identifies a new question for future research regarding potential functional roles for chemokine receptors selectively induced in apoptotic thymocytes. In order to study and quantify the presence of surface GAGs in live and apoptotic thymocytes, we analyzed by FACS the cell-binding activity of B18, a high affinity and specific GAG-binding protein expressed by some poxviruses [37]. As shown in Fig 4H, strong binding of B18 was detected on the surface of live thymocytes. However, interestingly, B18 binding was greatly impaired on apoptotic cells ($p < 0.001$) (Fig 4H). Consistent with this, we found that B18 binding was markedly reduced on UV-irradiated GAG-competent CHO-K1 cells, in this case, intriguingly, even before PS exposure on the cell plasma membrane (S7 Fig). Thus, apoptosis is accompanied by a severe depletion of cell surface GAGs. However, chemokines could still bind to these remaining low levels of GAGs on apoptotic thymocytes. To quantify the extent of GAG interference in our binding assays and to more definitively demonstrate the role of surface-exposed PS in chemokine binding to apoptotic cells, we tested and quantified the binding of CXCL11 to apoptotic thymocytes after treatment with proteinase K, an enzyme that eliminates surface GAGs (by cleaving the proteoglycan protein core that anchors GAGs to the cell surface) but not PS. As expected, proteinase K treatment completely abrogated B18 binding to live and apoptotic thymocytes; however, the binding of AnV to apoptotic cells was unaffected (Fig 4H). Importantly, like AnV, CXCL11 was still able to bind proteinase K–treated GAG-free apoptotic cells, whereas the treatment completely

eliminated CXCL11 binding to the surface of live thymocytes (Fig 4I). These results indicate that while GAGs entirely mediate CXCL11 binding to live thymocytes, they only play a partial role in apoptotic cells, with the other part of the binding most likely mediated by surface exposed PS. To directly test this hypothesis, we combined proteinase K treatment of thymocytes with PS-binding competition by unlabeled AnV. As shown in Fig 4J, both treatments, when applied independently, reduced the CXCL11 binding to apoptotic thymocytes by about 50% each. Importantly, consistent with the results in Fig 4F and 4G, competition with unlabeled AnV alone did not affect chemokine binding to live cells ($p = 0.275$), whereas proteinase K treatment alone sufficed to eliminate CXCL11 binding to live cells (Fig 4J). In contrast, the combination of both treatments completely abolished binding by this PS-binding chemokine to apoptotic thymocytes (Fig 4J). These results indicate that about half of CXCL11 binding to apoptotic cells involves GAGs, while the other half involves membrane-exposed PS. In summary, we conclude that PS-binding chemokines can bind to early apoptotic cells by a mechanism involving surface-exposed PS.

## Active endogenous phosphatidylserine-binding chemokines are transported on the surface of extracellular vesicles from apoptotic mouse thymus

We have demonstrated that many exogenous recombinant human chemokines are able to bind PS presented in pure form, liposomes, or on the surface of apoptotic blebs and apoptotic cells without affecting their bioactivity. To understand whether this interaction occurs in vivo and whether endogenous PS-binding chemokines can also be presented on the surface of endogenous extracellular vesicles to cognate receptors, we next analyzed chemokine-like activities triggered by apoptotic blebs produced in vivo in the thymus of C57BL/6j mice after intraperitoneal (i.p.) DEX injection.

Although we did not detect a high number or frequency of apoptotic thymocytes in DEX-treated mice, possibly due to rapid elimination by phagocytes, the high number of apoptotic blebs (SSC^lo) observed in these mice provided evidence for the efficacy of the treatment (panel A in S8 Fig). As expected, the apoptotic blebs exposed PS on the surface as indicated by AnV^+ staining (panel B in S8 Fig). First, we analyzed which chemokines were induced in the mouse thymus after DEX treatment. Using antibody-based arrays, we identified 6 chemokines with high basal or DEX-inducible expression levels in thymus extracts 6 hours and 18 hours after DEX injection: Ccl21, Ccl6, Ccl12, Cxcl10, Ccl9/10, and Ccl5 (Fig 5A). Of these, only Ccl12, Ccl21, and Cxcl10 bound to DOPS liposomes (Fig 5B). Therefore, these 3 chemokines were the main candidates to engage with PS on the surface of extracellular vesicles from mouse thymus. To test this, we next analyzed by western blot the presence of Ccl6, Ccl9/10, Ccl12, Ccl21, and Cxcl10 in different types of apoptotic blebs isolated from mouse thymus homogenates following a previously reported centrifugation protocol [38,39]. According to their size, apoptotic blebs can be classified as apoptotic bodies (ApoBDs, 1 to 5 µm) or apoptotic microvesicles (MVs, 0.1 to 1 µm) [40], which we separated from the final cleared SN using a series of differential centrifugation steps (Fig 5C). Although vesicle markers are known to largely overlap among different types of extracellular vesicles, we included mitochondrial voltage-dependent anion-selective channel 1 (VDAC1) and the tetraspanin CD81 as markers to identify the different fractions. Consistent with previous reports [39,41], VDAC1 and CD81 were more highly enriched in ApoBD and MVs, respectively (Fig 5D). On the other hand, as shown in Fig 5D, regardless of their PS-binding activity, all chemokines were found to some extent in the ApoBD and MV fractions as well as in the final SN fraction. However, it is important to note that chemokines might be transported encapsulated inside the vesicle or on the surface [42],

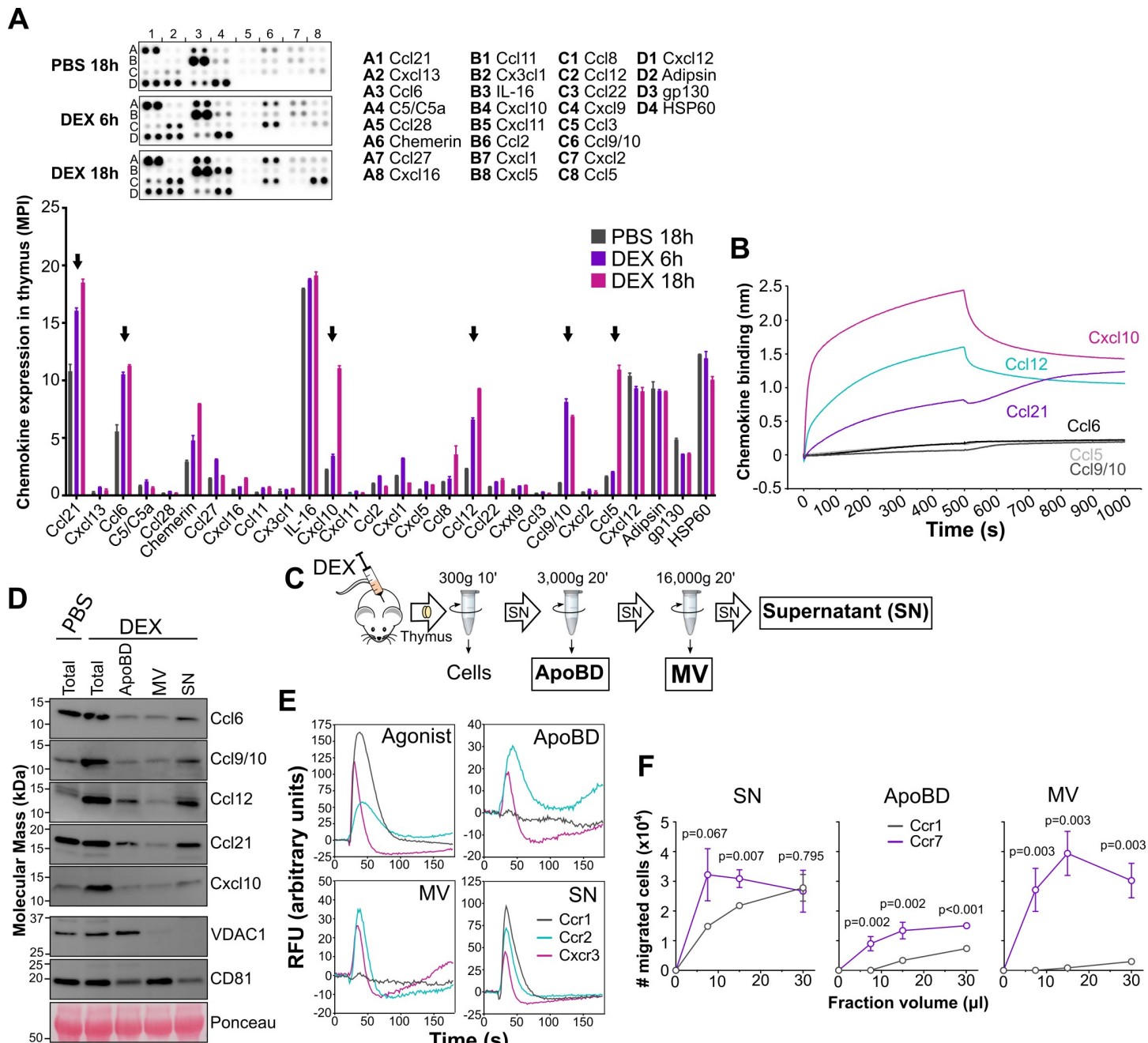

**Fig 5. Endogenous PS-binding chemokines presented on the surface of apoptotic blebs in vivo activate cognate GPCRs. (A)** Endogenous chemokine expression in the thymus of DEX-treated mice. Top, anti-chemokine antibody array membranes were incubated with 200 μg of total protein of mouse thymus extracts collected 6 hours or 18 hours after i.p. injection with PBS or DEX. The membranes are spotted in duplicate with capture antibodies for the chemokines indicated on the right side (position coordinates are indicated with lettered rows and numbered columns). Positions D2 (Adipsin), D3 (gp130), and D4 (HSP60) correspond to loading controls. Bottom, mean ± SD of the MPI for each chemokine. Black arrows point to highly DEX-induced chemokines. **(B)** BLI analysis of the binding of mouse Ccl5, Ccl6, Ccl9/10, Ccl12, Cxcl10, and Ccl21 (500 nM) to DOPS liposomes. Final binding sensorgrams were generated after subtraction of the binding recorded on DOPC sensors used as reference. **(C)** Schematic representation of the centrifugation steps followed to isolate ApoBDs, MVs, and the final cleared SN from mouse thymus homogenates. **(D)** Western blot analysis of Ccl6, Ccl9/10, Ccl12, Ccl21, and Cxcl10 and VDAC1 and CD81 as vesicle markers, in ApoBD, MV, and SN fractions isolated from thymus 18 hours after i.p. injection of mice with PBS or DEX. "Total" lane corresponds to the initial cell-free SN. Blot ponceau staining is shown as loading control. **(E, F)** Functional assays using ApoBD, MV, and SN fractions isolated from mouse thymus 18 hours after i.p. DEX injection. **(E)** Calcium flux assays using L1.2 reporter cell lines expressing Ccr1, Ccr2, and Cxcr3 (as indicated in the inset of the "SN" panel). The panel labeled "Agonist" shows the calcium flux response after addition of 50 nM of a known recombinant chemokine agonist for each receptor (Ccl9/10 for Ccr1, Ccl12 for Ccr2, and Cxcl10 for Cxcr3). Calcium flux recordings correspond to the mean of duplicates from one experiment representative of 3 independent experiments. **(F)** Chemotaxis assays using reporter cell lines expressing Ccr1 and Ccr7 (as indicated in the inset of the "ApoBD" panel) and increasing volumes (x-axis) of the indicated fractions. Data are the mean ± SD of the number of migrated cells of triplicates from one experiment representative of 3 independent experiments. *p*-Values from multiple *t* tests corrected for multiple comparisons by the Holm–Sidak method are indicated for each volume

data point. The underlying numerical values for the panels displaying summary numerical data can be found in S1 Data. ApoBD, apoptotic body; BLI, biolayer interferometry; DEX, dexamethasone; GPCR, G protein–coupled receptor; i.p., intraperitoneal; MPI, mean pixel intensity; MV, microvesicle; PS, phosphatidylserine; RFU, relative fluorescence units; SN, supernatant.

where, as we have shown before (Fig 4), they can engage with GAGs or exposed PS. Vesicle-encapsulated chemokine would be inaccessible to cellular receptors, and increasing evidence supports that, as we demonstrate here for CCL20 and CCL21 in Fig 3B, GAG binding is incompatible with simultaneous receptor activation for many chemokines [7]. In contrast, as we show in Fig 3, PS binding does not interfere with the chemokine bioactivity. Therefore, of the 3 chemokine forms transported by extracellular vesicles, the PS-bound form would be the most readily accessible to interact with and activate cognate receptors, and, hence, we reasoned that of the 6 chemokines of interest, only the PS-binding chemokines Ccl12, Ccl21, and Cxcl10 could functionally associate with the ApoBD and MV vesicle fractions. To explore this hypothesis, we tested the ability of the ApoBD, MV, and SN fractions to induce calcium flux responses in stable L1.2 cell lines expressing the cognate receptors for these 6 chemokines: Ccr2 for Ccl12, Ccr7 for Ccl21, Cxcr3 for Cxcl10, and Ccr1 for the 3 chemokines lacking PS-binding activity: Ccl5, Ccl6, and Ccl9/10 (Fig 5B). Of note, for these experiments, we decided to focus on fractions isolated from DEX-treated mice because, as shown in Fig 5A, the basal levels of some of the chemokines of interest were too low in PBS-inoculated mice to be reliably detected by functional assays (panel A in S9 Fig). As expected, an appropriate agonist control and the SN fraction triggered calcium flux responses in L1.2 cell lines expressing Ccr1, Ccr2, and Cxcr3 (Fig 5E). In contrast, ApoBD and MV vesicle fractions, each containing the same chemokines as SN (Fig 5D), activated calcium flux responses in Ccr2- and Cxcr3-expressing L1.2 cells but not in Ccr1-expressing cells (Fig 5E). Importantly, when the ApoBD fraction was filtered to remove the vesicles, the Ccr2-mediated calcium response was nearly eliminated (panel B in S9 Fig). Filtration of the SN fraction, however, did not affect activation of Ccr2-expressing cells (panel B in S9 Fig). Similarly, preincubation of the ApoBD fraction with a neutralizing anti-Ccl12 antibody, but not with an irrelevant control antibody, blunted the Ccr2-mediated calcium flux signal (panel C in S9 Fig). These results indicated that the ApoBD–Ccl12 complex is required for the activation of Ccr2 by the ApoBD fraction. To assess Ccl21 activity in the fractions, we used chemotaxis assays, since our Ccr7 reporter cell line responded poorly in calcium flux assays but well in chemotaxis assays to recombinant Ccl21 stimulation. As expected, SN induced chemotaxis of both Ccr1- and Ccr7-expressing cells (Fig 5F). However, only Ccr7-transfected cells migrated toward the ApoBD and MV fractions (Fig 5F). The reduced chemotactic activity observed with the ApoBD fraction compared to the MV fraction might be explained by the larger size of ApoBD, which could impede diffusion through the transwell filter and limit access to the reporter cells. Together, these results indicate that PS-binding chemokines, but not chemokines lacking PS-binding activity, on the surface of apoptotic blebs can be directly presented to cognate cellular receptors to activate cellular responses.

## Apoptotic bodies induce migration of phagocytes in a chemokine-dependent manner

We have shown that vesicle-bound PS-binding chemokines are capable of inducing cell migration of L1.2 cells. These findings open the interesting possibility that vesicle-bound chemokines might be responsible for the find-me signal activity attributed by several independent reports to the apoptotic blebs released by apoptotic cells [26–28]. To evaluate this hypothesis, we tested the ability of ApoBD, isolated from mouse thymus as in previous experiments, to induce phagocyte migration, the defining activity of find-me signals.

For these experiments, we used 3 types of relevant phagocytes, mouse bone marrow–derived macrophages (BMDMs) and 2 different lines of human monocytes, MonoMac-1 (MM1) and THP1. Of note, the latter has been used as a phagocyte model in several seminal papers of the apoptosis field to characterize the find-me signal activity of other molecules [21–23]. As shown in Fig 6A, ApoBD from DEX-treated mice induced robust migration of MM1 and THP1 monocytes comparable to that triggered by the cleared SN obtained from these mice. In contrast, the low monocyte migration induced by ApoBD and SN from PBS-inoculated mice did not statistically differ from the background migration toward assay media (Media) alone (Fig 6A), indicating that the thymus-derived monocyte chemotactic signal requires induction by proapoptotic DEX treatment. BMDM, however, displayed a different pattern of responsiveness to thymus-derived SN and ApoBD. Regardless of treatment, ApoBD induced a stronger BMDM migration than SN, which caused only a weak macrophage chemotaxis comparable to that observed with media alone (Fig 6B). These results reinforce the role of ApoBD as strong phagocyte-recruiting factors and suggest that unlike in the case of monocytes, functional levels of macrophage-recruiting agents are found in the thymus of PBS-inoculated mice. However, it is important to note that the ApoBD tested in these experiments were isolated from mouse thymus, an organ with high apoptotic rates even in naïve animals and where, as we show in Fig 5A, high basal levels of some chemokines can be detected. The inability of these chemokines expressed in the thymus of PBS-inoculated mice to cross-activate human chemokine receptors in MM1 and THP1 monocytes or the absence of their cognate receptors in these cells, may also explain why, unlike mouse BMDM, these human monocytes required DEX treatment to migrate in response to thymic ApoBD. In any case, together, these data confirm the chemotactic properties described by others for ApoBD, and to our knowledge, prove for the first time that ApoBD generated in vivo are chemotactic for phagocytes.

Unlike the functional experiments shown in Fig 5, where we assessed PS-bound chemokine activity using target cells transfected with a single chemokine receptor, phagocytes coexpress multiple endogenous chemokine receptors. For instance, we found that MM1 cells expressed detectable levels of CXCR4, CCR7, and CCR2 (Fig 6C), so that the cells might conceivably migrate in response to ApoBD by binding to multiple distinct vesicle-bound chemokines acting either independently or cooperatively. To test whether ApoBD-dependent phagocyte chemotaxis was mediated by chemokines, we assessed the effects of 2 multi-specific chemokine inhibitors, the viral chemokine-binding protein 35K (also termed "vCCI") and the viral chemokine homolog vCCL2 (also termed "vMIP-II"). The 35K protein is known to bind and block essentially all chemokines of the CC chemokine subfamily, whereas vCCL2 antagonizes many different chemokine receptors (CCR1, CCR2, CCR5, CCR8, XCR1, CX3CR1, and CXCR4) [43,44]. In addition, to study whether ATP, arguably the best characterized find-me signal, could be responsible for ApoBD-induced phagocyte migration in our system, we also tested the effect of apyrase, which hydrolyzes and eliminates ATP. As shown in Fig 6D, both chemokine inhibitors nearly completely neutralized the migration of MM1 and THP1 monocytes toward ApoBD from DEX-treated mice, whereas apyrase treatment was ineffective. These results indicate that chemokines, not ATP, mediate the phagocyte chemotactic activity of ApoBD isolated from mouse thymus in this system. It is important to note that this experiment was not designed to assess the well-established find-me signal activity of ATP and that our results do not question the proven role of nucleotides as phagocyte chemotactic molecules in other contexts. In addition, to confirm the specificity of these 3 inhibitors, we tested the ability of 35K, vCCL2, and apyrase to interfere with the migration of MM1 and THP1 in response to the recombinant chemokine Ccl12 (agonist for CCR2) and to purified ATP. As expected, 35K and vCCL2 blocked Ccl12- but not ATP-induced monocyte migration, and conversely, apyrase neutralized ATP but not the chemokine (S10 Fig). Finally, we found that almost 90%

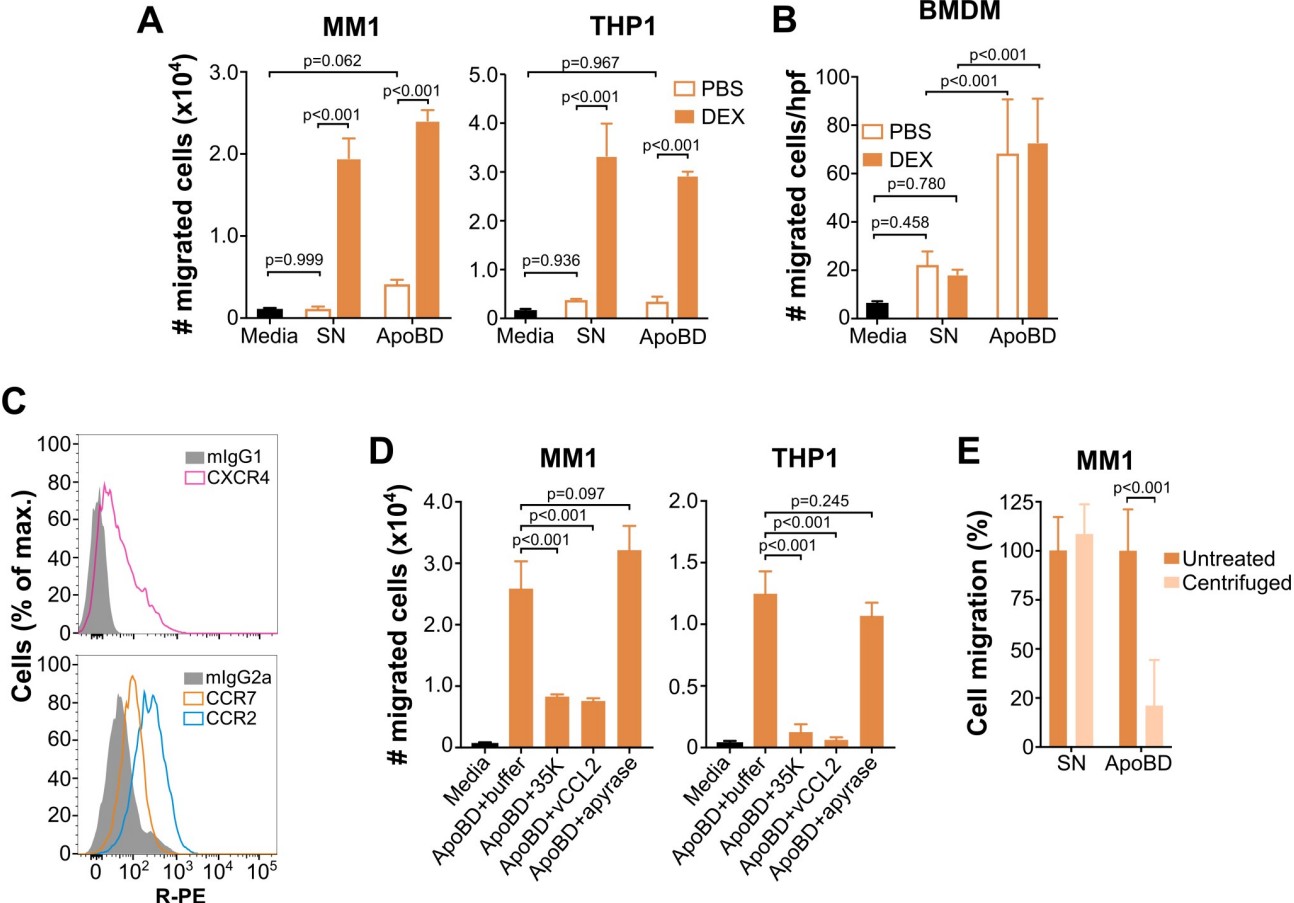

**Fig 6. Endogenous vesicle-bound chemokines mediate the chemotactic find-me signal activity of ApoBDs.** ApoBDs isolated from mouse thymus induce migration of monocytes (A) and macrophages (B). **(A)** Chemotaxis of the human monocytic lines MM1 and THP1 in response to media alone (black bar) or ApoBD and cleared SN isolated from thymus homogenates of C57BL/6j mice 18 hours after i.p. inoculation with PBS (open bars) or DEX (solid bars). Cell migration was analyzed in 8-μm pore size transwell plates for 2 hours at 37˚C. Bars represent mean ± SD migrated cells of triplicates from one experiment representative of 3 independent experiments. **(B)** Migration of BMDM in response to media alone (black bar) or ApoBD and SN isolated as in panel A from mice inoculated i.p. with PBS (open bars) or DEX (solid bars). Chemotaxis was assayed using 8-μm pore size polycarbonate membranes, and cells on the bottom side of the membrane were counted under the microscope after 1–2 hours at 37˚C. Migration was analyzed in triplicates, and 6 random hpf (400× magnification) were counted in each well. Bars represent mean ± SD cells/hpf from one experiment representative of 3 independent experiments. **(C)** Phagocytes express multiple chemokine receptors. FACS histograms showing the staining of MM1 cells with R-PE-conjugated antibodies for the receptors (open histograms) indicated in the insets or the appropriate isotype controls (gray). **(D)** The monocytic chemotactic activity of ApoBD is mediated by chemokines. Migration of MM1 and THP1 cells toward media alone (black bars) or ApoBD from DEX-inoculated mice preincubated with buffer, the ATP hydrolase apyrase (2 U/ml), or the broad-spectrum chemokine inhibitors 35K and vCCL2 (200 nM). Chemotaxis was analyzed as in panel A. Bars represent the mean ± SD number of migrated cells measured in triplicate from one experiment and are representative of results from 3 independent experiments. **(E)** Vesicles are required for the phagocyte chemotactic activity of ApoBD. Migration of MM1 monocytes was analyzed as in panel A in response to SN or ApoBD fractions, which were either untreated or centrifuged (16,000 x g, 45 minutes) to remove vesicles, as indicated in the legend. Data are the % migration relative to the migration observed with untreated SN or ApoBD samples. Bars represent the mean ± SD % cell migration of triplicate determinations from 2 experiments combined after subtracting the corresponding background migration observed with media alone in each experiment. All indicated *p*-values are from a 2-way (panels A, B, and E) or 1-way (panel D) ANOVA test with Tukey correction for multiple comparisons. The underlying numerical values for the panels displaying summary numerical data can be found in S1 Data. ApoBD, apoptotic body; BMDM, bone marrow–derived macrophage; DEX, dexamethasone; FACS, fluorescence-activated cell sorting; hpf, high-power field; i.p., intraperitoneal; R-PE, R-Phycoerythrin; SN, supernatant.

of the phagocyte migration activity of ApoBD was lost after depletion of the vesicles by high-speed centrifugation, whereas SN preserved its chemotactic activity entirely (Fig 6E). In summary, we conclude that ApoBD induce phagocyte migration through a mechanism mediated by vesicle-bound chemokines.

## Discussion

In this study, we identify a broadly shared capacity of chemokines to bind anionic phospholipids and provide proof of principle for PS-bound chemokines on extracellular vesicles as apoptotic find-me signals for phagocytes. PS binding was chemokine specific and high affinity, and it was observed when PS was presented either in pure form or in the context of liposomes and biological membranes from apoptotic cell lines, primary cells, and extracellular vesicles. Importantly, unlike GAG-bound chemokines, PS-bound chemokines retained their ability to simultaneously activate cognate GPCRs on leukocytes. Although our study began with a chemokine phospholipid screen, we have focused our attention on the interaction of chemokines with PS. Future work will be needed to analyze the biochemistry and biological significance of chemokine binding to other anionic phospholipids.

Since most chemokines are highly basic proteins, it is not surprising that they bind to highly anionic molecules like PS and GAGs. More surprising is the degree of specificity revealed by our biochemical screen, which suggested that chemokine–PS interaction is not exclusively charge driven. First, we showed that basic chemokines, such as CXCL8 (pI = 9.1), do not bind PS. Second, we found that most chemokines displayed a null/weak binding to phosphoinositides, which are more highly anionic phospholipids than PS. Furthermore, as demonstrated by BLI, many chemokine–PS interactions displayed a very slow dissociation rate, suggesting a contribution of hydrophobic or other stable bonds in the formation of the chemokine–PS complex. Importantly, not all GAG-binding chemokines were able to interact with PS. In particular, we showed that mouse and human CCL5, which are strong GAG-binders [32,45], do not bind PS. Furthermore, while soluble heparin neutralized CCL20 and CCL21, these chemokines remained fully active in the presence of very high doses of DOPS liposomes, indicating that unlike GAG binding, PS binding does not interfere with the chemokine chemotactic activity. Additionally, we showed that the elimination of CCL20–DOPS liposomes by pull-down reduced the overall chemotactic activity, which provides evidence that PS-bound chemokines may simultaneously activate cognate cellular receptors. These results suggest that the chemokine GAG- and PS-binding sites may require different molecular determinants and may map differently relative to the receptor-binding site. It is important to note that like GAGs, different chemokines may bind PS in different ways, and, therefore, case-by-case and comprehensive analyses may be required to fully understand the structural basis of chemokine–PS interaction.

We found that thymocytes undergoing apoptosis in vivo switch chemokine-tethering mechanisms, down-regulating anionic cell surface GAGs and up-regulating anionic PS exposed on apoptotic cells. Additional work will be needed to assess how general this phenomenon may be. GAGs act as substrates for the formation of haptotactic gradients promoting binding and oligomerization of chemokines on cell surfaces [5]. In fact, receptor-binding chemokine mutants unable to oligomerize or interact with GAGs are inactive in vivo [32]. Like GAGs, PS is mostly found on cell surfaces, and, here, we demonstrate that it can also induce chemokine oligomerization. Interestingly, we show that PS exposure is accompanied by a severe depletion of GAGs on the surface of apoptotic thymocytes, which bind chemokines in an AnV- and MFG-E8-susceptible manner. These 2 specific PS-binding proteins, however, did not affect chemokine binding to live GAG-competent thymocytes, which lack PS exposure on the cell surface. Furthermore, we demonstrate that while GAG removal by proteinase K treatment completely neutralized CXCL11 binding to live cells, this PS-binding chemokine was able to interact with proteinase K–treated apoptotic thymocytes. Importantly, CXCL11 binding to GAG-free apoptotic cells was neutralized by competition with AnV, demonstrating that PS-binding chemokines can exploit surface-exposed PS to interact with apoptotic cells. The fact that chemokines are capable of engaging with surface PS even in the presence of live cells,

which present abundant levels of surface GAGs, suggests that, despite the high affinity of some chemokine-GAG interactions, GAGs do not completely prevent PS-binding by chemokines. Therefore, we propose that as tissue apoptotic rates rise as a result of injury or infection, PS may be increasingly favored over GAGs as a mediator of chemokine oligomerization and haptotactic gradients. Furthermore, it is important to note that tumor cells and virally infected cells also expose PS [46–49]; therefore, chemokine modulation by PS may become more apparent in non-homeostatic conditions.

Effective and rapid apoptotic cell clearance is essential for the proper maintenance of homeostasis. Phagocytes use chemotactic find-me signals and the surface PS eat-me signal to reach and recognize apoptotic cells, respectively [50]. Surprisingly, to date, chemokines, arguably the most potent chemoattractants of the immune system, are only represented by CX3CL1 on the list of the most widely accepted find-me signals. Instead, several independent studies have reported that ApoBDs and MVs released by apoptotic cells can act as find-me signals that induce chemotaxis of phagocytes [26–28]. However, how these extracellular vesicles exert their chemotactic action had remained poorly understood. Here, we discovered that ApoBDs generated in vivo are potent chemoattractants for macrophages and monocytes and that this activity is mediated by vesicle-associated chemokines. Several lines of evidence provided in this study support the conclusion that PS-bound chemokine is the form responsible for the find-me activity of ApoBDs. First, GAG binding, unlike PS binding, impedes simultaneous interaction of the chemokine with cognate receptors. This is directly demonstrated here with CCL20 and CCL21 and supported by increasing chemokine structural evidence [5,7]. Second, thymic apoptotic blebs isolated from mouse thymus after proapoptotic treatment contain similar levels of Ccl12, Ccl21, Cxcl10, Ccl6, Ccl9/10, and Ccl5. Although all 6 of these chemokines can be expected to interact with GAGs, apoptotic blebs induced cellular responses in cells expressing the cognate receptors for PS-binding chemokines (Ccl12, Ccl21, and Cxcl10) but failed to activate Ccr1, the receptor for the 3 chemokines on this list lacking PS-binding activity (Ccl6, Ccl9/10, and Ccl5). Third, phagocyte migration induced by ApoBDs is neutralized by preincubation with the multispecific chemokine inhibitors, 35K and vCCL2, and by vesicle depletion through high-speed centrifugation. Taken together, we conclude that apoptotic blebs can act as find-me signals and attract phagocytes because they are coated with chemokines tethered to surface-exposed PS. The generation of genetically engineered mice expressing mutants of relevant chemokines lacking PS-binding activity but preserving their ability to interact with cellular receptors will be needed in future studies to establish the implications of this new chemokine PS-binding activity in phagocyte migration by apoptotic blebs, and therefore, in apoptotic cell clearance.

We propose a model whereby, in order to reach apoptotic cells, phagocytes would follow a trail of extracellular vesicles presenting chemokines on their surface (Fig 7). It is important to note that the cellular source of vesicle-bound chemokines remains to be identified. Apoptotic cells are known to produce phagocyte-attracting chemokines [51], and apoptotic blebs are certainly released by apoptotic cells, but soluble chemokines produced by neighboring healthy cells could still interact with the surface of apoptotic blebs (Fig 7). However, GAGs highly abundant on the surface of live cells might trap a significant fraction of these live cell-derived chemokines, whereas the lower GAG density on apoptotic cells might favor the interaction of apoptotic-cell derived chemokines with PS-exposing apoptotic blebs. It would also be interesting to understand how the vesicle–chemokine complex is processed by phagocytes on their way toward apoptotic cells and whether this complex might be eliminated by phagocytosis or chemokine receptor-mediated endocytosis (Fig 7), which would result in the elimination of inflammatory chemokines, contributing to the overall anti-inflammatory nature of apoptosis and collaborating with atypical chemokine receptors in the regulation of chemokine availability in the extracellular compartment [52]. Finally, different sets of chemokines might be

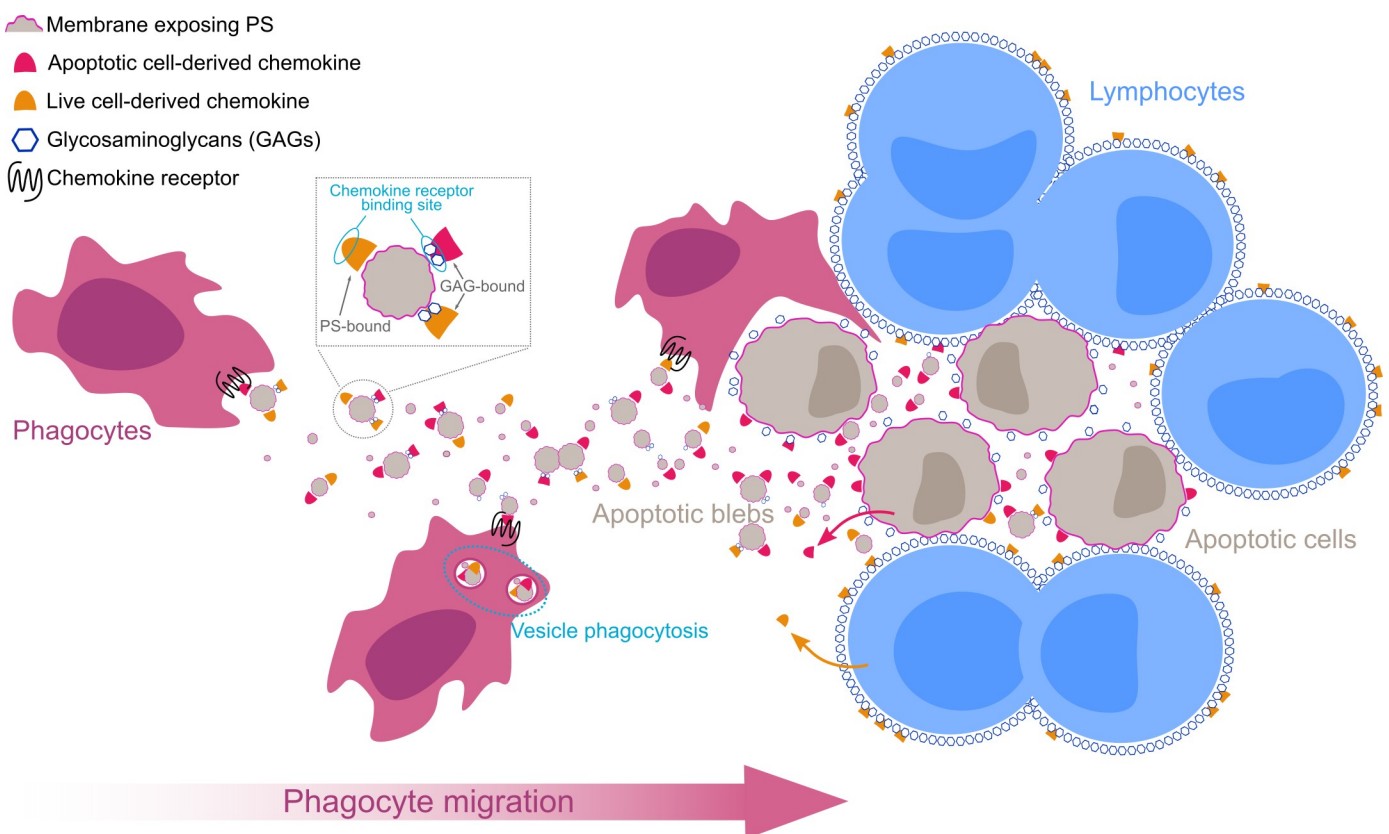

**Fig 7. Hypothetical model of chemokine-presenting apoptotic blebs guiding phagocytes toward apoptotic cells for their elimination.** Graphical representation of a proposed model for the role of vesicle-bound chemokines in apoptotic cell clearance. Chemokines produced by live or apoptotic lymphocytes interact with the surface of apoptotic blebs released from apoptotic cells via GAGs or membrane-exposed PS. GAGs mask the receptor-binding site of the chemokine, whereas PS-bound chemokines are able to simultaneously bind to and activate cognate chemokine receptors expressed by phagocytes. A gradient of chemokine-presenting apoptotic blebs guides phagocytes toward apoptotic cells. Phagocytes engulf and eliminate chemokine-presenting apoptotic blebs and apoptotic cells reducing inflammation. GAG, glycosaminoglycan; PS, phosphatidylserine.

induced during different modes of cell death (necrosis, necroptosis, apoptosis, and pyroptosis) and in different tissues, which could result in the attraction of different subsets of phagocytes. A comprehensive analysis of the chemokines and chemokine receptors expressed by dying cells and phagocytes, respectively, would help to identify chemokine-bleb-receptor axes involved in the recruitment of phagocytes and to understand their influence on the kinetics and efficiency of dying cell clearance.

Here, we chose to validate the biological relevance of chemokine–PS interactions in the context of apoptosis. However, many PS-binding chemokines identified by our study are not known to recruit phagocytes, and, therefore, they are unlikely to impact apoptotic cell clearance. Instead, PS binding may regulate the activity of chemokines that attract non-phagocytic cell types in other biological processes where cell membrane PS becomes exposed and accessible for the interaction with soluble cytokines, for example, in cancer and during infectious diseases. Extracellular vesicles like ApoBDs, MVs, and exosomes have received increasing attention in recent years as essential elements of intercellular communication in cancer and immunity [53,54]. RNA transcripts, microRNA, enzymes, cytokines, chemokines, and cellular receptors, among others, are transported from cell to cell by extracellular vesicles [55]. Research in the field has focused principally on encapsulated nucleic acids and integral membrane proteins of the vesicle. This might be a consequence of the fact that while it is easy to

understand how cell state can be modulated by RNAs, cellular receptors or cytosolic enzymes delivered by extracellular vesicles, how vesicle-encapsulated cytokines and chemokines, whose action depends on the interaction with receptors in the plasma membrane of a target cell, might contribute to vesicle-mediated cell-to-cell communication is less apparent. In this regard, our study provides proof of principle that signaling cytokines can be displayed and distributed in vivo attached to the surface of extracellular vesicles. Tumor cells and viruses might hijack this system to distribute cytokines and activate or antagonize cellular receptors in distant organs. Understanding how lipids interact with secreted proteins may help explain the functional roles played by extracellular vesicles during homeostasis, cancer, and infection.

## Materials and methods

### Animals

C57BL/6j mice were obtained from The Jackson Laboratory (Bar Harbor, Maine, United States of America). All mice were maintained under specific pathogen-free housing conditions at an American Association for the Accreditation of Laboratory Animal Care–accredited animal facility at the National Institute of Allergy and Infectious Diseases (NIAID) and housed in accordance with the procedures outlined in the Guide for the Care and Use of Laboratory Animals under the protocol LMI-8E approved on December 31, 2015 and annually renewed by the Animal Care and Use Committee of NIAID.

### Cells

CHO-K1 and the GAG-deficient variant CHO-745 cells (generous gift from Dr. Antonio Alcami, Spanish Research Council, Spain) were grown in DMEM/F12-Glutamax (Life Technologies, Carlsbad, California, USA) supplemented with 10% FBS. MonoMac 1 (MM1) (kind gift from Dr. Antonio Alcami, Spanish Research Council, Spain) and THP1 (ATCC, Manassas, Virginia, USA) monocytic cell lines were grown in RPMI-Glutamax media (Life Technologies) supplemented with 10% FBS and also 55 μM 2-mercaptoethanol in the case of THP1 cells. L1.2 cells (kindly provided by Dr. Eugene Butcher, Stanford University, California, USA) were grown in RPMI-Glutamax supplemented with 10% FBS, 1 mM sodium pyruvate, and 0.1 mM nonessential amino acids. All cells were cultured at 37°C and 5% $CO_2$. Suspension Expi293F cells were maintained in FBS-free Expi293 Expression Medium (both from Life Technologies) and grown at 37°C, 8% $CO_2$ with constant agitation at 125 rpm. BMDM were generated from frozen stocks of C57BL/6j bone marrow cells cultured at 37°C and 5% $CO_2$ in RPMI-Glutamax media supplemented with 20% FBS and 30% L929 cell-conditioned media as previously described [56].

### Reagents

Untagged human/mouse recombinant chemokines produced in *Escherichia coli* were obtained from Peprotech (Rocky Hill, New Jersey, USA). Of note, recombinant CXCL16 included only the extracellular domain of human CXCL16. The synthetic biotinylated human chemokines CCL2, CCL3, CCL11, CXCL11, and CCL20 were purchased from Almac (Craigavon, United Kingdom). All primary anti-chemokine antibodies used in this study were purified polyclonal rabbit antibodies from Peprotech. Recombinant human MFG-E8 protein produced in mouse myeloma cells was purchased from R&D Systems (Minneapolis, Minnesota, USA).

### Generation of stable cell lines expressing chemokine receptors

Cell lines expressing the mouse chemokine receptors Ccr1, Ccr2, Ccr6, Ccr7, and Cxcr3 were generated in L1.2 cells. cDNA clones in pCMV6 plasmid for these chemokine receptors were

obtained from Origene (Rockville, Maryland, USA). L1.2 cells ($2 \times 10^6$ cells) were transfected with 2 μg of plasmid using the SG Cell Line transfection kit and a 4D-Nucleofector X (both from Lonza, Walkersville, Maryland, USA) following the manufacturer's instructions. Moreover, 48 hours post-transfection, cells were seeded in 24-well plates in pools containing 2,400 cells/well in selection media (RPMI-Glutamax, 10% FBS, 1 mM sodium pyruvate, 0.1 mM nonessential amino acids and 1 mg/ml geneticin). After 7 days at 37˚C, media was replaced with fresh selection media, and the cells were incubated for 7 additional days. Then, surviving pools were screened for the expression of the corresponding chemokine receptor by calcium flux assays. Pools displaying the strongest calcium flux responses were further selected by limiting dilution in 96-well plates. Receptor-expressing clones were selected by calcium flux assays. Highly responsive clones were expanded and frozen in geneticin-free selection media containing 5% DMSO. For subsequent experiments, cell lines were grown in selection media.

## Lipid array binding assays

The lipid-binding specificity of recombinant human chemokines (Peprotech) was determined using Membrane Lipid Strips obtained from Echelon Biosciences (Salt Lake City, Utah, USA). These strips are spotted with 100 pmol of 15 different lipids found in cell membranes. First, strips were incubated with blocking buffer (1X Tris-buffered saline [TBS], 0.1% Tween-20, 3% BSA) for 1 hour at room temperature. Then, recombinant chemokines were added at 0.1 μg/ml in blocking buffer and incubated for 1 hour at room temperature. Strips were washed with TBS-T (TBS, 0.1% Tween-20), and bound protein was detected with specific rabbit anti-chemokine antibodies (Peprotech) followed by an HRP-conjugated anti-rabbit antibody (Abcam, Cambridge, Massachusetts, USA). Strips were developed with WesternBright Sirius (Advansta, San Jose, California, USA) and imaged in an Omega Lum C imager (Gel Company, San Francisco, California, USA).

## Liposomes

The phospholipids DOPC, DOPS, 1,2-dioleoyl-sn-glycero-3-phosphoethanolamine-N-(cap biotinyl) (DOPEbiot), and 1,2-distearoyl-sn-glycero-3-phosphoethanolamine-N-[biotinyl (polyethylene glycol)-2000] (DSPE-PEGbiot) were all purchased from Avanti Polar Lipids (Alabaster, Alabama, USA). In this study, 2 different types of liposomes were used: "DOPC," composed of DOPC:DOPEbiot at a 95:5 ratio (weight %), and "DOPS," composed of DOPS: DOPC:DOPEbiot at a 30:65:5 (weight %). Phospholipids resuspended in chloroform were mixed at the indicated ratios (1-mg total lipid), and the solvent was evaporated in a SpeedVac Concentrator (Fisher Scientific, Pittsburgh, Pennsylvania, USA). Dried lipids were rehydrated in 0.5 ml of PBS or TBS for 2 hours at room temperature. Subsequently, large unilamellar vesicles were prepared by extrusion ($>$ 11 passes) through a 0.1-μm membrane at room temperature using a mini-extruder (Avanti Polar Lipids). Liposomes were used immediately after preparation.

## Liposome binding assays

The ability of chemokines to interact with phospholipid liposomes was tested by ELISA and BLI.

For ELISA-based binding experiments, DOPC and DOPS liposomes prepared in TBS were immobilized at 10 μg/ml in binding buffer (TBS, 0.1% BSA) on streptavidin-coated high capacity plates (Thermo Fisher Scientific, Waltham, Massachusetts, USA). Increasing concentrations (0.01 to 10 μg/ml) of recombinant chemokines in binding buffer were incubated in the wells for 15 minutes at room temperature. Plates were washed extensively with TBS, and

bound chemokine was detected with specific rabbit anti-chemokine antibodies (Peprotech) and HRP-conjugated anti-rabbit secondary antibody development (Abcam). The (Twin)-Strep-tag /Strep-Tactin affinity purification system (IBA Lifesciences, Göttingen, Germany) was used to produce twin-strep-tagged fusion chemokines which were detected with an HRP-conjugated anti-Streptag mAb (IBA Lifesciences). Plates were developed with TMB One Component (Surmodics, Eden Prairie, Minnesota, USA), and the reaction was stopped with sulfuric acid before measuring the absorbance at 450 nm ($A_{450}$) in a FlexStation 3 microplate reader (Molecular Devices, Sunnyvale, California, USA). Nonspecific binding to the plate was corrected by the subtraction of the $A_{450}$ recorded for each chemokine concentration in wells incubated without liposome. Where indicated, chemokine binding to liposome-coated wells was competed by preincubation of chemokines with increasing concentrations of lipids for 30 minutes at room temperature.

BLI experiments were performed in an Octet RED384 system (Pall ForteBio, Fremont, California, USA) essentially as previously described with some modifications [57]. For these experiments, liposomes were prepared in PBS, and DSPE-PEGbiot instead of DOPEbiot was used as biotinylated phospholipid to improve the immobilization of the liposomes on the biosensors. Briefly, SA biosensors (Pall ForteBio) pre-wetted in PBS for at least 10 minutes were equilibrated in PBS for 60 seconds. Then, DOPC and DOPS liposomes were immobilized to a final 2- to 4-nm or 0.5- to 1-nm response for binding screening or kinetic purposes, respectively. Subsequently, biosensor tips were washed for 60 seconds in PBS and blocked for 300 seconds with 0.05% BSA in PBS. Finally, biosensors were washed for 150 seconds in PBS, and the association of recombinant chemokines was recorded for 500 seconds followed by a 500 seconds dissociation in PBS. All the steps were performed at 30°C and 1,000 rpm. For binding screening, chemokines were tested at 1 μM. For kinetic analysis, increasing concentrations (50 nM to 1,000 nM) of chemokine were used. Binding to DOPC liposomes was used as reference and subtracted from the binding to DOPS liposomes using Octet Data Analysis software (Pall ForteBio). For determination of binding kinetic parameters ($K_{on}$, $K_{off}$, and $K_D$), sensorgrams were adjusted to a 1:1 Langmuir model using Octet Data Analysis software (Pall ForteBio).

## Chemokine–phospholipid cross-linking assays

The capacity of different natural phospholipids—brain PE, brain PC, brain PS, and heart CL, all from Avanti Polar lipids—to induce chemokine oligomerization was studied by cross-linking experiments. First, phospholipids stored in chloroform at 10 mg/ml were diluted 1:1,000 in 20 mM HEPES. Then, 50 ng of chemokine were incubated for 30 minutes at room temperature in the presence or absence of phospholipid at a 1:8 (chemokine:total lipid) molar ratio in 12 μl of 20 mM HEPES. Then, 3 μl of the cross-linker $BS_3$ (Life Technologies) were added for a 0.25-mM final concentration, and samples were incubated at room temperature for additional 30 minutes. The reaction was stopped by the addition of 5 μl of 4X SDS-PAGE loading buffer, and chemokine oligomerization was analyzed by western blot. Chemokine oligomers were detected by specific rabbit anti-chemokine antibodies (Peprotech) followed by development with an HRP-conjugated anti-rabbit antibody (Abcam).

## Chemokine binding to apoptotic cells

Apoptotic CHO-745 cells and mouse thymocytes were used for chemokine binding experiments. Cell death of GAG-deficient CHO-745 cells was induced by irradiation with UV light. Cells cultured on 100 mm plates were washed once with PBS, and cell monolayers were irradiated with 100 mJ in a UV-Stratalinker 2400 (StrataGene, San Diego, California, USA) with the plate lid off or on to generate the mock control. Six hours after treatment, cells were collected

with trypsin-EDTA followed by at least 2 washes with culture media to remove the EDTA, which may interfere with the $Ca^{2+}$-dependent binding of AnV. Apoptotic thymocytes were generated by incubation of thymocytes ($10^7$ cells/ml) freshly isolated from C57BL/6j mice with 1 μM DEX (Sigma, Saint Louis, MO, USA) for 4 to 5 hours in RPMI-Glutamax supplemented with 10% FBS. All binding assays were performed in AnV Binding Buffer (BioLegend, San Diego, California, USA). For competition experiments, cells were preincubated for 5 minutes at room temperature with 1 μg of the unlabeled PS-binding proteins MFG-E8 (R&D Systems) and AnV (BioLegend). Also, where indicated, thymocytes were treated beforehand with 0.5 mg/ml of proteinase K (Roche, Indianapolis, Indiana, USA) in PBS at 37˚C for 20 minutes to remove cell surface GAGs. Proteinase K was inactivated with 2 mM phenylmethylsulfonyl fluoride (PMSF) in PBS supplemented with 3% BSA incubated 5 minutes at room temperature before thymocytes were used for binding assays. Moreover, 3 to $5 \times 10^5$ cells were incubated with 500 to 250 nM biotinylated CCL2, CCL3, CCL11, CCL20, CXCL11 (all from Almac, Craigavon), or AnV (BioLegend) in 100 μl of binding buffer for 10 minutes at room temperature. Then, cells were pelleted by centrifugation (1,200 rpm, 5 minutes) and incubated in 100 μl of binding buffer containing 0.125 μg of streptavindin-APC (BioLegend) for 10 minutes at room temperature. Finally, cells were resuspended in binding buffer and stained with 3.5 μg/ml PI (BioLegend), and, in the case of the thymocytes, also with 0.1 μM YO-PRO (Life Technologies). Events were acquired in an LSR Fortessa cytometer, and FlowJo software (both from BD Bioscience, San Jose, California, USA) was used for data analysis.

## Analysis of surface GAGs on mouse thymocytes

The levels of GAGs on the surface of mouse thymocytes were determined and quantified by analyzing the cell-binding activity of the specific GAG-binding viral protein B18 (kind gift from Dr. Antonio Alcami, Spanish Research Council, Spain). Apoptosis of mouse thymocytes from C57BL/6j mice was induced by incubation with 1 μM DEX (Sigma) for 4 to 5 hours in RPMI-Glutamax supplemented with 10% FBS. Where indicated, after apoptosis induction, cell surface GAGs were removed by cell treatment with proteinase K as explained above. Then, $5 \times 10^5$ thymocytes were incubated for 20 minutes on ice with his-tagged B18 protein (200 nM) in PBS-staining buffer (1x PBS supplemented with 1% BSA and 1% FBS). Cell surface-bound B18 was detected by staining with a rabbit anti-His tag mAb (clone D3I1O, Cell Signaling Technology, Beverly, Massachusetts, USA) followed by an anti-rabbit IgG F(ab')$_2$ fragment conjugated with Alexa Fluor 488 (Cell Signaling Technology) in PBS-staining buffer. Cells were washed with AnV binding buffer and stained with an APC-conjugated AnV and PI (both from BioLegend). B18 binding to live (AnV$^-$ PI$^-$) and apoptotic thymocytes (AnV$^+$ PI$^-$) was analyzed in a LSR Fortessa cytometer and subsequent analysis using FlowJo (both from BD Bioscience).

## Isolation of extracellular vesicles from mouse thymus

Apoptosis and apoptotic blebs production were induced in vivo in C57BL/6j mice by i.p. injection of 250 μg of DEX (Sigma) in PBS. Thymocytes were isolated from thymi of mice inoculated with DEX or PBS alone 6 hours or 18 hours after injection by filtering through a 70-μm cell strainer rinsed with 500 μl/thymus of the assay buffer appropriate for the subsequent application: For calcium flux assays, samples were prepared in 1:1 vol:vol culture media:calcium buffer (RPMI-Glutamax 10% FBS:HEPES (20 mM) in HBSS); for chemotaxis assays, samples were prepared in RPMI-Glutamax supplemented with 10 mM HEPES and 0.5% BSA. Apoptosis and apoptotic blebs induction was confirmed by AnV/PI staining and FACS analysis. The different vesicle fractions and the final cleared SN were isolated from these thymus

homogenates by a series of centrifugation steps as illustrated in Fig 5C. First, cells were removed by centrifugation at 300 x g and 4˚C for 10 minutes. Complete cell removal in the resulting SN was confirmed by flow cytometry before centrifugation at 3,000 x g and 4˚C for 20 minutes to pellet ApoBDs. The resulting pellet was resuspended in 500 μl/thymus of the appropriate buffer and labeled ApoBD, and the SN was centrifuged at 16,000 x g and 4˚C for 20 minutes to extract the MVs. This last pellet was resuspended in 500 μl/thymus of the appropriate buffer and labeled MVs, and the final cleared SN was labeled SN. Chemokine-like activity in ApoBD, MV, and SN fractions was assayed by chemotaxis and calcium flux assays (see below) ideally immediately or within 4 days after isolation storing the samples at 4˚C. Also, where indicated, vesicular matter in SN and ApoBD fractions was depleted by centrifugation at 16,000 x g and 4˚C for 45 minutes. Of note, typically, thymi from 2 to 3 mice per treatment group (PBS or DEX) were pooled together for vesicle and SN isolation, which resulted in ApoBD and MVs pellet volumes of approximately 5 to 15 μl. Therefore, these pellets were extensively diluted (at least 100-fold) after resuspension with the indicated volumes of the appropriate buffer.

## Chemokine expression in mouse thymus

Expression levels of thymic chemokines in thymus extracts or in ApoBD, MVs, and SN fractions purified from C57BL/6j mice after i.p. injection of PBS alone or with 250 μg of DEX (Sigma) were quantified using Proteome Profiler Mouse Array kit (R&D Systems) or analyzed by western blot, respectively. For chemokine quantification, mouse thymus was collected 6 hours or 18 hours after treatment and homogenized in 1 ml of cold PBS containing cOmplete protease inhibitor cocktail (Roche) using a tissue homogenizer (Omni International, Kennesaw, Georgia, USA). Protein concentration was determined by BCA assay, and array membranes were incubated with an extract volume corresponding to 200 μg of protein. Arrays were developed following the manufacturer's instructions, and membranes for all samples were imaged simultaneously in an Omega LumC imager (Gel Company, San Francisco, California, USA). The mean pixel intensity for each signal was analyzed using ImageJ. To analyze the presence of several chemokines of interest in thymic vesicle fractions, 15 μl of the ApoBD, MVs, SN samples, or the initial cell-free SN (Total) were analyzed by SDS-PAGE and immunoblot with appropriate rabbit anti-chemokine pAb (all from Peprotech) and an HRP-conjugated anti-rabbit secondary antibody (Abcam). Blots were developed using WesternBright Sirius HRP substrate (Advansta, San Jose, California, USA) and imaged in an Omega LumC imager (Gel Company) at exposing times optimized to visualize the chemokine bands in the vesicle fractions.

## Chemotaxis assays

Chemotaxis of L1.2 stable cell lines and the human monocytes MM1 and THP1 was analyzed using 96-well ChemoTx plates (Neuroprobe, Gaithersburg, Maryland, USA) with 5 μm or 8 μm pore size filters, respectively. Cells were washed twice with chemotaxis assay buffer (RPMI-Glutamax supplemented with 10 mM HEPES and 0.5% BSA), and $2 \times 10^5$ cells/well resuspended in assay buffer were placed on top of the filter. Media alone, purified ATP (Sigma), recombinant chemokines, or the thymic fractions ApoBD, MVs, and SN were added in assay buffer to the bottom wells as chemotactic stimuli. Chemokines and ATP were used at the indicated concentrations and, except where specified, fractions were used without further dilution. In some experiments, chemokines and thymic fractions were preincubated for 30 minutes at room temperature with soluble heparin, liposomes (DOPC or DOPS), apyrase (New England Biolabs, Ipswich, Massachusetts, USA) or the chemokine inhibitors 35K/vCCI and vCCL2/vMIP-II (both from R&D Systems) at the indicated concentrations in assay buffer. Since RPMI media contains high levels of free biotin, DMEM-Glutamax supplemented with

0.1% of BSA and 10 mM HEPES was used as assay buffer in experiments where liposomes–chemokine complexes were quantified in streptavidin-immobilized ELISA plates or pulled down with Strep-Tactin-coupled beads. Total migrated L1.2 cells in the bottom well after a 3 to 4 hours incubation at 37˚C and 5% CO2 were calculated by the addition of 5 μl/ well of CellTiter 96 Aqueous One Solution (Promega, Madison, Wisconsin, USA) and interpolation of the $A_{490}$, measured using a FlexStation 3 microplate reader (Molecular Devices) following the manufacturer's recommendations, in a standard cell curve. Migrated MM1 and THP1 cells after 2 to 3 hours were counted in the bottom well under the microscope by the trypan blue exclusion method.

Migration of BMDM toward media alone or the thymic ApoBD and SN fractions was assayed in a 48-well microchamber (Neuroprobe) using 8-μm pore size polycarbonate membranes. BMDM washed and resuspended in chemotaxis assay buffer were placed at 50,000 cells/well in the top wells. After 1 to 2 hours at 37˚C and 5% CO2, membranes were fixed and stained with Hoechst 33342 (1.6 μM), and cells migrated to the bottom side of the membrane were counted in an Axiovert 200 inverted fluorescence microscope (Zeiss, White Plains, New York, USA).

### Calcium flux assays

L1.2 stable lines were seeded at 300,000 cells/well in black clear-bottom 96-well plates and incubated with FLIPR Calcium 6 dye (Molecular Devices) for 2 hours at 37˚C following the manufacturer's recommendations. As mentioned above, thymic ApoBD, MVs, and SN fractions used in calcium flux assays were prepared in calcium assay buffer (1:1 vol:vol RPMI-Glutamax 10% FBS:HEPES (20 mM) in HBSS) and used without further dilution. Calcium flux signals were recorded for 180 seconds in a FlexStation 3 plate reader (Molecular Devices) using the instrument's built-in pipettor to automatically dispense 25 μl of buffer, the appropriate recombinant chemokine agonists (50 nM in assay buffer), or thymic fractions at 20 seconds after the initiation of the recording.

## Supporting information

**S1 Fig. BLI assays can be used to analyze the PS-binding activity of recombinant proteins.** Binding of recombinant MFG-E8 (200 nM) to BLI biosensors immobilized with DOPC (blue sensorgram) or DOPS (pink sensorgram) liposomes. BLI, biolayer interferometry; MFG-E8, milk fat globule-epidermal growth factor 8; PS, phosphatidylserine.
(PDF)

**S2 Fig. Some primary anti-chemokine antibodies may not recognize their target chemokine when in complex with DOPS liposomes.** Primary antibodies used in ELISA or protein–lipid overlay assays to detect CCL21, CXCL9, CXCL6, or CXCL3 were immobilized onto BLI amine-reactive biosensors. BLI binding sensorgrams for the interaction of DOPS liposomes alone (green) and 400 nM of each chemokine alone (magenta) or preincubated with 0.5 mg/ ml of DOPS (yellow) or DOPC (blue) liposomes are shown. Increase in the binding response in the presence of liposomes is indicative of the binding of a large analyte (chemokine–liposome complex). BLI, biolayer interferometry.
(PDF)

**S3 Fig. Chemokines detect 5%–10% of PS in liposomes.** BLI experiments showing the binding of the indicated chemokines (500 nM) to DOPS liposomes containing decreasing amounts of PS (as indicated in the inset of the CCL19 graph). Binding to DOPC liposomes was subtracted from all binding curves. BLI, biolayer interferometry; PS, phosphatidylserine.
(PDF)

**S4 Fig. Chemokine–DOPS liposome complexes are chemotactically active. (A)** Liposomes do not induce chemotaxis in the absence of chemokine. Cell migration of Ccr1-, Ccr6-, and Ccr7-expressing L1.2 cells (y-axis) in the presence of the same concentrations (x-axis) of DOPS or DOPC liposomes used in Fig 3A but without chemokine was assayed in transwell plates for 3–4 hours at 37˚C. Media alone (0:0, chemokine:lipid molar ratio) and 1 nM of the appropriate chemokine agonist (as indicated on the left side of each graph) in the absence of liposome (1:0, chemokine:lipid molar ratio) were included as negative and positive controls, respectively. Results from controls (Cntrl.) and cells stimulated with DOPS or DOPC liposomes are separated by vertical dashed lines and labeled above the top graph. Bars represent the mean ± SD of triplicate determinations from one experiment representative of 2 independent experiments. **(B)** Pull-down of CCL20–DOPS liposome complexes decreases their availability in solution. CCL20 (1 nM) was incubated with buffer or a $10^4$-fold molar excess of DOPC or DOPS liposomes doped with a small amount of biotinylated DOPE. A total of 50 μl of the liposome suspension before (input) and after (output) pull-down with 30 μl of Strep-Tactin beads were analyzed in triplicate by ELISA in streptavidin-coated plates. Liposome-bound CCL20 was detected with a rabbit anti-CCL20 polyclonal Ab followed by an HRP-conjugated anti-rabbit antibody, and the $A_{450}$ was determined after development with TMB One Component solution. Bars represent the mean ± SD of data from one experiment representative of 2 independent experiments. The *p*-value from a 2-tailed *t* test for the analysis of CCL20 + DOPS input vs. output is indicated. **(C)** Depletion of CCL20–DOPS liposome complexes by pull-down reduces cell migration. CCL3 or CCL20 (as indicated above each graph, 1 nM) were incubated with buffer or a $10^4$-fold molar excess of DOPC or DOPS liposomes. Then, chemokine–liposome complexes were pulled down with Strep-Tactin-beads, and the chemokine activity remaining in the SNs was tested by chemotaxis assays using L1.2 cell lines expressing the appropriate chemokine receptor (Ccr6 for CCL20 and Ccr1 for CCL3). Bars represent mean ± SD of the number of migrated cells in triplicate determinations from one experiment representative of 3 independent experiments. *p*-Values are from 1-way ANOVA test with Bonferroni correction for multiple comparisons. The underlying numerical values for the panels displaying summary numerical data can be found in S1 Data. SN, supernatant. (PDF)

**S5 Fig. Glycosylated recombinant chemokines interact with PS and necrotic cells. (A)** Purification of recombinant CCL3 expressed in Expi293F cells. Coomassie-stained acrylamide gel showing the purification steps of human CCL3. Human CCL3, CXCL9, and CCL21 were tagged with a twin-strep tag (tst) at the carboxyl terminus and expressed in Expi293F cells. Recombinant proteins were purified from cell SNs by affinity chromatography using Strep-Tactin XT columns. **(B)** CXCL9tst and CCL21tst interact with PS-containing liposomes. The binding of in-house produced glycosylated recombinant chemokines to DOPC or DOPS liposomes was analyzed by ELISA. Increasing doses (x-axis) of the different chemokines (as indicated in the inset of the right panel) were incubated in wells immobilized with DOPC (left panel) or DOPS (right panel) liposomes. Wells were washed extensively with TBS, then bound chemokine was detected with an HRP-conjugated anti-tst mAb, and the $A_{450}$ was determined after developing with TMB One Solution substrate. Data are the mean ± SD of triplicates from one experiment representative of 3 independent experiments. **(C)** PS-binding chemokines interact with the surface of dying CHO-745 cells. Immunofluorescence images show the binding of CCL3tst, CXCL9tst, and CCL21tst (as indicated on the left side of each row) to UV-irradiated CHO-745 cells. Cells cultured on coverslips were exposed to 100 mJ of UV light using a Stratalinker. Six hours after treatment, coverslips were incubated with 400 nM of each chemokine in AnV binding buffer. After washing, samples were stained with Strep-Tactin XT

conjugated with DY-488, then fixed and mounted using Prolong Gold with DAPI. Epifluorescence images were acquired in a Zeiss Anxiovert 200M inverted microscope (200× magnification). DAPI (first column), Strep-Tactin staining (second column), and merge images (third column) are shown. White arrows in the DAPI panels point at examples of dying cells displaying fragmented and condensed nuclei. In the merge panels, DAPI and Strep-Tactin staining are shown in cyan and magenta, respectively, as indicated on the bottom right corner. A white scale bar corresponding to 50 μm is shown in the merge CCL21tst panel. The underlying numerical values for the panels displaying summary numerical data can be found in S1 Data. PS, phosphatidylserine; SN, supernatant; UV, ultraviolet.
(PDF)

**S6 Fig. Expression of chemokine receptors in apoptotic CHO-745 cells and apoptotic mouse thymocytes.** The expression of the cellular receptors for the chemokines included in the cell-binding assays shown in Fig 4 was analyzed by FACS in CHO-745 cells (A) and mouse thymocytes (B) as indicated above each panel. **(A)** Cxcr3, Ccr3, and Ccr7 are not expressed in live or apoptotic CHO-745 cells. Mock-treated or UV-irradiated CHO-745 cells (as indicated above the graph columns) were stained with AnV-APC and R-PE conjugated antibodies for the chemokine receptors Ccr3, Ccr7, and Cxcr3 (as indicated on the y-axis of the corresponding graphs). Chemokine receptor-AnV dot plots are shown. Dot plots for the staining with the pertinent isotype controls are shown above the corresponding columns. **(B)** Ccr3, and to a lower extent, Ccr2 and Ccr6, but not Cxcr3, are expressed specifically by apoptotic thymocytes. Freshly isolated mouse thymocytes were incubated with 1 μM DEX at 37˚C. After 4 hours, thymocytes were stained with AnV-APC and PE-conjugated anti-Ccr2, anti-Ccr3, anti-Ccr6, or anti-Cxcr3 antibodies as indicated on the y-axis of the corresponding graphs. Chemokine receptor-AnV dot plots are shown below the corresponding isotype control for each anti-chemokine receptor antibody. In both panels A and B, numbers indicate the % of events in each gate. AnV, annexin V; DEX, dexamethasone; R-PE, R-Phycoerythrin; UV, ultraviolet.
(PDF)

**S7 Fig. UV-irradiated CHO-K1 cells display highly reduced levels of cell surface GAGs.** CHO-K1 (GAG competent) and CHO-745 (GAG deficient) cells were exposed to 100 mJ of UV-light. Six hours after irradiation, cells were collected, and the cell binding of the GAG-binding B18-His protein was analyzed by FACS with an anti-His mAb followed by an anti-mouse Alexa Fluor 488-conjugated antibody. Before the analysis, cells were stained with PI and APC-conjugated AnV. Top dot plots show the PI and AnV staining of mock- and UV-irradiated CHO-K1 or CHO-745 cells as indicated above each graph. Numbers correspond to % of events in each gate. Bottom histogram graph shows the binding of B18-His to AnV$^-$ PI$^-$ populations (color coded in the top dot plots) from mock- (orange) or UV-irradiated CHO-K1 (green) and CHO-745 cells (purple) as indicated in the legend on the right side of the graph. Staining of CHO-745 cells in the absence of B18-His (PBS, gray) is shown as reference. AnV, annexin V; PI, propidium iodide; UV, ultraviolet.
(PDF)

**S8 Fig. Injection of DEX induces the production of PS-containing apoptotic blebs in mouse thymus. (A)** C57BL/6j mice were i.p. injected with PBS or DEX, and thymocytes were isolated 6 hours or 18 hours after treatment (as indicated on the right side of each graph row) and stained with PI and APC-conjugated AnV. In the left column, SSC-FSC dot plots showing gates for the cells (Cells, blue) and apoptotic blebs (SSC$^{lo}$, black). In the right column, PI-AnV dot plots of the events from the "Cells" gate of each condition. Numbers indicate the % of events in each gate. **(B)** Histograms of the AnV staining of live cells (AnV$^-$ PI$^-$) and apoptotic

blebs (SSC<sup>lo</sup>, black). AnV, annexin V; DEX, dexamethasone; FSC, forward scatter; i.p., intra-peritoneal; PI, propidium iodide; PS, phosphatidylserine; SSC, side scatter.
(PDF)

**S9 Fig. ApoBD–Ccl12 complex is required for Ccr2 activation by ApoBD fraction. (A)** DEX treatment is required for strong Ccr2 activation by SN fractions isolated from mouse thymus homogenates. Calcium flux of Ccr2-expressing L1.2 cells in response to thymic SN isolated from PBS- (black) or DEX-inoculated (pink) mice. **(B)** Calcium flux response for buffer alone (black) and for the mouse SN and ApoBD fractions previously filtered (pink) or not (blue) through a 0.2 μm filter. **(C)** Calcium flux response obtained with the ApoBD fraction preincubated or not (No Ab, pink) with 10 μg/ml of an anti-Ccl12 antibody (orange) or a control anti-Ccl6 antibody (blue). All SN and ApoBD fractions were isolated 18 hours after mouse treatment. Calcium recordings correspond to the mean of duplicates from one experiment representative of 2 independent experiments. The underlying numerical values for the panels displaying summary numerical data can be found in S1 Data. ApoBD, apoptotic body; DEX, dexamethasone; RFU, relative fluorescence units; SN, supernatant.
(PDF)

**S10 Fig. Apyrase and recombinant 35K and vCCL2 proteins specifically inhibit ATP- and chemokine-mediated cell migration, respectively.** Chemotaxis assays proving the chemokine or ATP inhibitory specificity of 35K and vCCL2, or apyrase, respectively. Migration of MM1 and THP1 cells induced by Ccl12 (1 nM) or ATP (500 nM) preincubated with buffer, 200 nM of the chemokine inhibitors 35K or vCCL2, or 2 U/ml of apyrase was analyzed as in A. Bars represent mean ± SD number of migrated cells measured in triplicate from one experiment representative of 3 independent experiments. $p$-Values are from a 2-way ANOVA test with Tukey correction for multiple comparisons. The underlying numerical values for the panels displaying summary numerical data can be found in S1 Data.
(PDF)

**S1 Table. Binding (second column) of human chemokines (first column) at 1 μM to DOPS liposomes by BLI.** Binding fold change relative to the binding of the known PS-binding chemokine CXCL16 is indicated in the third column. Binding responses <0.1 nm are highlighted in orange. Fold change values >2.0 and <0.5 are highlighted in blue or purple, respectively. BLI, biolayer interferometry; PS, phosphatidylserine.
(PDF)

**S1 File. Supporting Materials and methods.**
(PDF)

**S1 Data. Spreadsheet containing individual sheets of the underlying numerical data for panels in Figs 1 and 3–6 and S4, S5, S9 and S10 Figs.**
(XLSX)

**S1 Raw Images for Gels and Blots. Raw uncropped images of western blot and dot plot membranes of Figs 2 and 5.**
(PDF)

## Acknowledgments

We thank Dr. Mark Connors and Dr. Kenta Matsuda from the HIV-Specific Immunity Section of the Laboratory of Immunoregulation at the National Institute of Allergy and Infectious Diseases (Bethesda, Maryland, USA) for granting us access to their Octet RED384 BLI system.

## Author Contributions

**Conceptualization:** Sergio M. Pontejo, Philip M. Murphy.

**Formal analysis:** Sergio M. Pontejo.

**Investigation:** Sergio M. Pontejo.

**Writing – original draft:** Sergio M. Pontejo, Philip M. Murphy.

**Writing – review & editing:** Sergio M. Pontejo, Philip M. Murphy.

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
