## [Editor Report · Decision Letter 0]

11 May 2020

Dear Dr Pontejo, 

Thank you very much for submitting your manuscript entitled "Chemokines as phosphatidylserine-bound ‘find-me’ signals in apoptotic cell clearance" for consideration as a Research Article by PLOS Biology. 

We appreciated your patience when my colleagues and I have assessed your manuscript and consulted with an Academic Editor who is an expert in this area. While we found the topic of your study interesting and relevant, we would need data that demonstrate that the binding of the chemokines to the apoptotic cells is specifically via PS, by having proper negative controls (manipulating PS levels available on the cell surface somehow, with kd of enzymes, activating scramblases independent of apoptosis, etc.) and data that rule out the role of proteins in the binding (by kd, ko or protease digestion of the cell surface). We don't find these data in the current submission, and thus regrettably have to decline the submission in its current form.

While we cannot consider your manuscript further for publication in PLOS Biology, we suggest, as an alternative, that you consider transferring this manuscript to PLOS ONE (http://journals.plos.org/plosone/). 

PLOS ONE is a peer-reviewed journal that accepts scientifically sound primary research. The review process at PLOS ONE differs from other PLOS journals in that it does not judge the perceived impact of the work or whether this falls within a particular area of research. Rather, it focuses on whether the study has been performed and reported to high scientific and ethical standards, and whether the data support the conclusions. This approach helps to eliminate the rejection cycles that authors commonly encounter when submitting to one journal after another. Please note that the journals are editorially independent and we therefore cannot guarantee the outcome if you choose to pursue a transfer.

If you would like to submit your work to PLOS ONE, please click the following link:

<DeepLinkData><DeepLinkTypeID>27</DeepLinkTypeID><peopleID>1441103</peopleID><userSecurityID>984eb7f1-7272-41f6-bbc4-9671dcc6dc77</userSecurityID><documentID>41468</documentID><revision>0</revision><manuscriptNumber>PBIOLOGY-D-20-01298</manuscriptNumber><docSecurityID>7077cfa5-b237-457a-a4f4-371745134ab0</docSecurityID></DeepLinkData>

If you do NOT wish to submit your work to PLOS ONE, please click this link to decline: 

<DeepLinkData><DeepLinkTypeID>28</DeepLinkTypeID><peopleID>1441103</peopleID><userSecurityID>984eb7f1-7272-41f6-bbc4-9671dcc6dc77</userSecurityID><documentID>41468</documentID><revision>0</revision><manuscriptNumber>PBIOLOGY-D-20-01298</manuscriptNumber><docSecurityID>7077cfa5-b237-457a-a4f4-371745134ab0</docSecurityID></DeepLinkData>

Should you choose to transfer your submission to PLOS ONE, you will receive a confirmation email within 24-48 hours of accepting the transfer. Please note, all PLOS journals are editorially independent and vary in submission requirements. Your submission details and manuscript files will be transferred automatically; once in the PLOS ONE Editorial Manager site, your submission will be returned to you and you will be asked to provide additional information before you can finalize your new submission to PLOS ONE. If you have any questions, please feel free to contact the journal at plosone@plos.org.

Thank you for giving us the opportunity to consider your work.

Sincerely,

Di Jiang, PhD

Associate Editor

PLOS Biology

---

## [Decision Letter · Decision Letter 1]

24 Jun 2020

Dear Dr Pontejo,

Thank you very much for submitting your manuscript "Chemokines as phosphatidylserine-bound ‘find-me’ signals in apoptotic cell clearance" for consideration as a Research Article at PLOS Biology. Your manuscript has been evaluated by the PLOS Biology editors, an Academic Editor with relevant expertise, and by four independent reviewers.

In light of the reviews (below), we will not be able to accept the current version of the manuscript, but we would welcome re-submission of a much-revised version that takes into account the reviewers' comments. We will need to see that the concerns from reviewer 1 be fully and satisfactorily addressed, as they are very valid. In addition, our Academic Editor raises a related point that PS is exposed not only by apoptotic cells, which release anti-inflammatory signals, but also by cells dead by necroptosis and other forms of regulated necrosis, which are inflammatory forms of cell death. It is common that, in vitro, cells that start dying by apoptosis then undergo secondary necrosis because they cannot be removed as in the tissue. You will need to address this concern in your revision. We cannot make any decision about publication until we have seen the revised manuscript and your response to the reviewers' comments. Your revised manuscript is also likely to be sent for further evaluation by the reviewers.

We expect to receive your revised manuscript within 2 months. 

**IMPORTANT - SUBMITTING YOUR REVISION**

*Re-submission Checklist*

*Published Peer Review*

*PLOS Data Policy*

*Blot and Gel Data Policy*

Sincerely,

Di Jiang, PhD

Senior Editor

PLOS Biology

REVIEWS:

Reviewer #1: Previously, Shimaoka et al (reference 10) showed that CXCL16 binds phosphatidylserine (PS). In this manuscript, the authors claim that many other chemokines bind phosphatidylserine (PS). This is probably interesting finding. However, the manuscript is not well prepared, and the results are not consistent to each other, and are not convincing. The physiological role of this finding is also not clear. My detailed comments are as follow. 

Materials and Methods: How each chemokine was prepared? Are they native proteins or recombinant proteins? If they are recombinant, are they produced in mammalian cells or E. coli? If they are commercially available non-glycosylated form (Line 283), is it prepared in E. coli. Chemokines carry many cysteine residues. How the authors know the molecules are fully active. Some chemokines are membrane-bound form. Only the extracellular region was used? The authors use antibodies to recognize chemokines. Please describe sources of each antibody.

Fig. 1A: The authors use the respective antibodies to detect the chemokines in the array and ELISA assay, which make the comparison between chemokines very difficult. The authors have the system to prepare the strep tagged chemokines (Fig. S4). If the authors want to compare the ability of each chemokine to bind PS, all chemokines should be prepared in the same way and should be detected by the same detection system.

Fig. 1A, 1B/1D, and 1E/1F. Some chemokines bind to PS in one assay (array), but not others (ELISA or BLI). While other chemokines do not bind PS in the array, but binds PS in the other assay. It is very confusing, and indicates that something wrong in the assay or the experimental system. 

Fig. 1E and 1F: I am not sure whether this BLI analysis is giving the correct data or not. Why the response values do not go down to the bottom in the washing step? In some cases (CCL-21, CXCL4 and CXCL9), they even increase. Why? I do not think that the fitting to Langmuir model gives the correct kinetic values under these conditions?

Fig. S1: How MFG-E8 is prepared? Please show the Kd for the interaction of MFG-E8 with PS.

Fig. 2: PS-induced oligomerization is interesting. Please describe the experimental procedure in more detail. Phospholipids are usually insoluble in water, and are dissolved in CHCl3 and methanol. How phospholipids were mixed with chemokines in this experiment? The Mr of phospholipids are about 1,000. When the authors say that chemokine and phospholipids are mixed at 1:8, the 50 ng of chemokines were mixed with 50 ng of phospholipids? What is the mechanism of the oligomerization? Or, how phospholipids cause the oligomerization of chemokines? 

Fig. 3: With the CXCL16 molecules that carry mutation in the conserved receptor-binding domain (chemokine-domain), Shimaoka, T. et al., (reference 15) showed that chemokines (at lease CXCL16) binds to oxidized LDL via the receptor binding domain. Here, the authors claim that the PS-engaged chemokines can have the chemo-attractive activity. To strength this conclusion, a mutational analysis is necessary. Or, please show the mutants that cannot bind PS still have the chemo attractive activity. 

Line 274, Fig. 4A: PI+ AnV+ cells are NOT apoptotic cells. PI+ means that cells underwent necrosis with ruptured plasma membranes. Please see Crowley et al. (CSHL Protoc, doi: 10.1101/pdb.prot087288).

Reference 28 should be replaced by Hanayama et al. (Nature 417, 182, 2002).

Reviewer #2: The authors investigated in depth the interaction of phosphatidylserine (PS) and chemokines. They propose that binding of chemokines to PS on apoptotic cells contributes to the 'eat-me' signal for phagocytes to eliminate the dying cell. The authors attempt to solve the underlying mechanisms, since chemokine presentation on cells per se is not an 'eat-me' signal, rather functions as guidance cue for immune cells during inflammation. Chemokine activity (receptor interaction) is not affected by binding to PS, albeit PS induces chemokine aggregation, much in contrast to heparin and GAG binding. Finally, they propose that exosomes released from apoptotic cells decorated with chemokines from chemotactic cues to attract phagocytes.

This is an interesting and conceptually novel observation, which extends our knowledge on the chemokine system. All experiments are conclusive and build a logical line to support the hypothesis. I have only minor suggestions to complete the manuscript.

Comments:

Figure 3: at least for one chemokine a dose response curve (CK) in the presence of different DOPS/DOPC ratios should be given.

Figure 4: The effect of UV irradiation (panel A, right column AnV/PI) should be quantified. The quadrants should indicate % of cells. Panel B: should indicate the absolute cell numbers. From the current presentation it is difficult to appreciate the increase in chemokine binding after UV irradiation. Panel E needs proper statistical analysis.

Line 335 4E should read 4F 

Reviewer #3: In this manuscript, Pontejo and Murphy describe studies demonstrating functional chemokine binding by phosphatidylserine and its potential role in apoptotic cell clearance. This is a very interesting, and well-written, manuscript which provides important new insights into aspects of in vivo chemokine cellular presentation and function. Overall I am enthusiastic about this paper but I have a number of concerns many of which are more philosophical in nature and can possibly be dealt with by an enhanced discussion.

I am not persuaded that the data clearly indicate a role in apoptotic cell clearance as stated in the title. Certainly this is an attractive model but many of the chemokines that are highlighted as having strong PS binding (e.g. CCL20, CCL21 CXCL11 etc) would not be expected to attract phagocytes and therefore their role as PS bound 'eat me' signals seems unlikely. I wonder whether there is a broader biology here that might be worth considering. 

Another point is that, in terms of attracting phagocytes to the 'eat me signals' this would imply a gradient of sorts from the apoptotic cell to the phagocyte. This then raises the question of where the PS-bound chemokines are coming from. If they are coming from another source then it is unlikely that their binding by apoptotic cell surface, and microvesicle, PS would allow a gradient to form that would attract phagocytes to the dying cell. In contrast, if the chemokines are actually produced by the dying cell then this might be a very attractive model and allow the apoptotic cell to be the chemokine 'source' and not a 'sink'. As apoptotic cells are likely to be producing inflammatory chemokines, including attractants for monocytes and macrophages (see figure 5A), this may give much more specificity to this axis in terms of attracting phagocytes (although I accept that many of these chemokines are not good PS binders). I suspect that it would be very difficult to formally prove this idea and so I am not asking for experiments to be done in this regard but this at least needs to be discussed.

In terms of broader biology is also possible that the exposure of PS on the surface of apoptotic cells, and on apoptotic microvesicles, could act in a similar way to circulating ACKR1 i.e. as a chemokine 'sponge' mopping up chemokines and contributing to resolution of inflammatory and immune responses. This may help further explain the strong binding of non-phagocyte-attracting chemokines and certainly is worth discussing in the text.

Overall, therefore, this is an interesting paper but I am not persuaded that the results clearly indicate a role in apoptotic cell clearance as suggested by the title.

Other points. 

Statistical analysis should be improved throughout. The analysis of figures 4D, 5F and supplementary figure 3A in particular requires proper statistical testing and indication of significance levels. These figures are fundamental to the conclusions being drawn and therefore this analysis is essential.

---

## [Decision Letter · Decision Letter 2]

14 Oct 2020

Dear Dr Pontejo,

Thank you very much for submitting a revised version of your manuscript entitled "Chemokines as phosphatidylserine-bound ‘find-me’ signals in apoptotic cell clearance" for consideration as a Research Article at PLOS Biology. Thank you also for your patience and please accept again my sincere apologies for the long delay in sending you the decision. This revised version of your manuscript has been evaluated by the PLOS Biology editors, the Academic Editor and by three of the original reviewers. As I mentioned earlier, Reviewer 1, who was an expert in apoptosis, didn't provide any comments we could share with you. Thus we decided to contact an arbitrating reviewer to assess that part of the manuscript.

As you will see, three of the original reviewers are now satisfied with the manuscript, however Reviewer 5, who has looked at the previous comments of Reviewer 1, is unconvinced of the physiological relevance of the findings and thinks the experiments do not demonstrate at this stage a role for chemokines bound to PS in apoptotic cell clearance. The reviewer suggests some experiments that can be performed to address this and to confirm the model proposed. After discussing these comments with the academic editor and the rest of the team, we have decided to give you another chance to address the concerns that remain unaddressed.

In light of the reviews (attached below), we will not be able to accept the current version of the manuscript, but we would like to would welcome re-submission of a revised version that takes into account Reviewer 5's comments. We cannot make any decision about publication until we have seen the revised manuscript and your response to the reviewers' comments. Your revised manuscript is also likely to be sent for further evaluation by the reviewers.

We expect to receive your revised manuscript within 3 months. 

**IMPORTANT - SUBMITTING YOUR REVISION**

*Re-submission Checklist*

*Published Peer Review*

*PLOS Data Policy*

*Blot and Gel Data Policy*

Sincerely,

Ines

--

Ines Alvarez-Garcia, PhD,

Senior Editor,

ialvarez-garcia@plos.org,

PLOS Biology

Rev. 2:

The authors have answered all my concerns. No alterations requested. Given that migrating leukocytes internalize chemokines during chemotaxis, it would be interesting in the future to reveal whether chemokines dissociate from liposomes/apoptotic blebs or are internalized with the lipid vesicles.

Rev. 3:

All comments have been satisfactorily addressed. I am still a little uncomfortable with the title but the biology is of sufficient interest that I will not request any further changes.

Rev. 4:

The authors have done a nice job of responding to reviewer concerns with appropriate changes to the manuscript. The inclusion of the model (Fig 6) is a nice addition. I recommend publication.

Rev. 5

While the binding of chemokines (other than CXCL16) to PS interesting, the physiological relevance of these findings is questionable. Indeed, and in contrast with the title of the manuscript, the authors do not provide any results demonstrating a role in apoptotic cell clearance. The results therefore appear very preliminary for the claims made by the authors.

Additional comments:

- Figure 2. It is not clear why the authors used a cross linker, as it might artefactually aggregate the chemokines.

- Figure 3. What is the effect of PS or PC alone on migration?

- Figure 4. A-C, The authors claim binding of the chemokines to GAG-deficient late apoptotic CHO cells, but PI+/AnV+ cells are necrotic and not apoptotic. These CHO cells may – or may not - originate from late apoptotic cells undergoing secondary necrosis. In order to know, shorter kinetics of UV treatment should be performed. The authors also refer to apoptotic blebs, but do not characterize them. Figure 4B, The absence of increase binding in response to UV treatment certainly questions the notion of preferential binding to PS exposed by dead cells. Figure 4C, the authors should do a better characterization of apoptotic blebs. SSC on log scale, size beads, ImageStream is standard analysis to demonstrate vesicular nature of blebs (not just debris on the cytometer): PI low?? Figure 4E, The results show much stronger binding to live thymocytes than to early apoptotic ones, which does not really fit with the proposed model. The authors try to explain it by the levels of GAG expression in live cells, but how does this solve the physiological problem? Where would specificity in the response come from then? Also, not clear what the statistical analysis refer to or means. The legend states “Bars represent mean ± SD of duplicates from one experiment representative of 3 independent experiments”!! Also, in order to demonstrate GAG- and cognate receptor-independent binding of the chemokines to early apoptotic cells, the authors should have performed binding assays with GAG-deficient cells KO for the respective receptor. Figure 5. D, markers should be used by WB to differentiate the factions. CCl2 does not seem to be induced, in contrast to the 2x increase reported in A. Figure 4E-F, similar experiments should have been performed out of the PBS injected control mice. Figure 4F, L1,2 cells?? These are not a phagocytes – model migration but do not model the complex interaction of a macrophage/DC/phagocyte with an apoptotic target. Figure 6. Not really a summary of the findings but rather a speculative model. The authors represent the chemokine associated to PS in apoptotic blebs recognized by phagocytes as originating from apoptotic cells, but the cellular source of chemokines is not at all demonstrated

Discussion

Line 502: highly atypical chemotactic cues… what about nucelotides? Metabolites? Negates a lot of research on the nature of find me signaling.

Line 500: “these apoptotic blebs only activate cognate GPCRs of PS-binding chemokines

Of course...” because this is in their in vitro models using transduced GPCRs on B cells .

Line 517: live cell-derived chemokines might be expressed at lower levels and might not attract phagocytes. Highly speculative!

Line 530: the complex of thymus apoptotic blebs with mouse Ccl12 is required to activate mouse Ccr2-expressing cells. ‘required’ for B cell activation!

General comment

The authors have overstated their conclusions based on the data present in the paper. A role for PS-binding chemokines in apoptotic cell clearance is purely speculative based on their data. Critically, the authors should address the capacity for PS-bound chemokines to modulate migration of phagocytes (which will interact with an apoptotic body or cell through the engagement of multiple surface receptors, namely receptors for PS and opsonins), or indeed to affect efferocytosis by phagocytes.

---

## [Decision Letter · Decision Letter 3]

7 Apr 2021

Dear Dr Pontejo,

Thank you for submitting your revised Research Article entitled "Chemokines as phosphatidylserine-bound ‘find-me’ signals in apoptotic cell clearance" for publication in PLOS Biology. I have now obtained advice from one of the original reviewers and have discussed their comments with the Academic Editor. 

Based on the reviews, we will probably accept this manuscript for publication, provided you satisfactorily address the remaining points raised by the reviewer (see below). Please also make sure to address all the data and other policy-related requests.

We would also like you to consider a suggestion to improve the title:

"Chemokines act as phosphatidylserine-bound ‘find-me’ signals in apoptotic cell clearance"

We expect to receive your revised manuscript within two weeks. 

*Published Peer Review History*

*Early Version*

Sincerely,

Ines

--

Ines Alvarez-Garcia, PhD,

Senior Editor,

PLOS Biology

DATA POLICY:

Thank you for including a data file with the raw data underlying some of the graphs. We do need, however, all individual quantitative observations that underlie the data summarized in the figures. Please provide the individual numerical values that underlie the summary data displayed in the following figure panels as they are essential for readers to assess your analysis and to reproduce it:

Fig. 1B, F; Fig. 4A-D, F, H, I; Fig. 5B; Fig. 6C; Fig. S1; Fig. S2; Fig. S3; Fig. S6A, B; Fig. S7 and Fig. S8

In addition, we found several errors in the file you provided - please clarify/amend them accordingly:

Fig. 1C values seem to be mislabelled as B

Fig. 1D values seem to be mislabelled as C

Fig. 1E values seem to be mislabelled as D, but some values shown in the graphs are missing in the data file (eg. CCL2, CCL13, CCL22, CCL23, CXCL1, CXCL2, CXL4, …)

Fig. 4H values seem to be mislabelled as G or missing.

Please also note that for figures containing FACS data, we ask that you provide FCS files and a picture showing the successive plots and gates that were applied to the FCS files to generate the figure. We are aware that these files can be quite large, so if that is the case, we would suggest you to deposit them in the FlowRepository (http://flowrepository.org/) and please make sure that the data are made publicly avalable at this time.

Please ensure that figure legends in your manuscript include information on WHERE THE UNDERLYING DATA CAN BE FOUND, and ensure your supplemental data file/s has a legend.

BLURB

Please also provide a blurb which (if accepted) will be included in our weekly and monthly Electronic Table of Contents, sent out to readers of PLOS Biology, and may be used to promote your article in social media. The blurb should be about 30-40 words long and is subject to editorial changes. It should, without exaggeration, entice people to read your manuscript. It should not be redundant with the title and should not contain acronyms or abbreviations. For examples, view our author guidelines: https://journals.plos.org/plosbiology/s/revising-your-manuscript#loc-blurb

Reviewers' comments

Rev. 5:

Firstly, I thank the authors for the efforts they have gone to in order to improve the manuscript in response the critical analysis by us and other reviewers. The rewording and new data shown has indeed improved clarity and provided more convincing support for their thesis of PS-binding by chemokines, and a putative role in phagocyte recruitment to clear apoptotic cells.

Notably, the use of macrophage/monocytes to study migration; and the use of B18 binding and proteinase K to demonstrate the relative contribution of GAG versus PS-binding to chemokine retention on the surface of dying cells.

This reviewer remains skeptical about the real in vivo relevance of the chemokine-PS interaction in the context of apoptotic cell clearance.

Nonetheless, I do agree that the experiments are for the most part thoroughly performed and the findings presented in this manuscript are of sufficient interest to justify publication.

As the authors discuss, PS-binding by such a range of chemokines may indeed be pertinent to physiological/pathological processes outside the context of cell death and clearance.

I have some small (but critical) suggestions for publication:

Figure 4

"E) Quantification of the median fluorescence intensity (MFI) of the binding of the indicated biotinylated chemokines ... Bars represent mean ± SD MFI of duplicates. Results are from one experiment representative of 3 independent experiments. p values from multiple t tests with Holm-Sidak correction for multiple comparisons are indicated."

>> Please confirm that statistics were performed on experimental replicates (greater than or equal to 3) and NOT technical duplicates. Showing S.D. for technical duplicates is incorrect.

Line 687 "this activity is vesicle-dependent"

>>Use of centrifugation to deplete the vesicles (and presumably their associated chemokines) is a valid approach; but does not necessarily demonstrate that the chemotactic activity of these cytokines is dependent on vesicle integrity. Use of a surfactant would demonstrate that. Given the authors do not fully characterize their apoptotic vesicles/blebs beyond low SSC (ie to prove vesicle integrity) I would advocate removal of the reference to 'vesicle-dependent' activity.

Line 543: "BMDM, however, displayed a different pattern of responsiveness to thymus-derived SN and ApoBD…" and Line 546: "These results reinforce the role of ApoBD as strong phagocyte-recruiting factors and indicate that unlike in the case of monocytes, functional levels of macrophage-recruiting agents …"

>> The authors should mention briefly the considerations of using mouse-derived chemokines to induce cellular responses mediated by human receptors. Moreover, the difference in receptor expression (and their cross species ligand recognition) on the monocyte/macrophages used may explain the differential responsiveness to the ApoBD derived from PBS or DEX injected mice?

>>Although an admirable attempt, the use of apyrase in an ex vivo context does not seem to be an entirely pertinent experiment. That is, ATP release is a potent chemoattractant acting over a short distance from dying cell target to phagocyte. Yet in a context in which the ApoBD have been isolated from the target, one would intuit that the relative contribution of ATP to their chemotactic effect is, of course, negligible. Figure 6E could be moved to supplemental data.

>>Right column in figure 3A - label the title (eg 'binding to liposomes')

---

## [Editor Report · Decision Letter 4]

5 May 2021

Dear Dr Pontejo,

On behalf of my colleagues and the Academic Editor, Ana Garcia-Saez, I am pleased to say that we can in principle offer to publish your Research Article entitled "Chemokines act as phosphatidylserine-bound ‘find-me’ signals in apoptotic cell clearance" in PLOS Biology, provided you address any remaining formatting and reporting issues. These will be detailed in an email that will follow this letter and that you will usually receive within 2-3 business days, during which time no action is required from you. Please note that we will not be able to formally accept your manuscript and schedule it for publication until you have made the required changes.

PRESS

Thank you again for supporting Open Access publishing. We look forward to publishing your paper in PLOS Biology. 

Sincerely, 

Ines

--

Ines Alvarez-Garcia, PhD 

Senior Editor 

PLOS Biology